# Open-Pit Granite Mining Area Extraction Using UAV Aerial Images and the Novel GIPNet

Xiaoliang Meng, Ding Zhang, Sijun Dong and Chunjing Yao *

School of Remote Sensing and Information Engineering, Wuhan University, Wuhan 430079, China; xmeng@whu.edu.cn (X.M.); frankzhang@whu.edu.cn (D.Z.); dyzy41@whu.edu.cn (S.D.)

* Correspondence: yaocj@whu.edu.cn; Tel.: +86-13517212891

**Abstract:** The ability to rapidly and accurately delineate open-pit granite mining areas is pivotal for effective production planning and environmental impact assessment. Over the years, advancements in remote sensing techniques, including the utilization of satellite imagery, LiDAR technology and unmanned aerial vehicles, have revolutionized the way mining areas are monitored and managed. Simultaneously, in the context of the open-pit mining area extraction task, deep learning-based automatic recognition is gradually replacing manual visual interpretation. Leveraging the potential of unmanned aerial vehicles (UAVs) for real-time, low-risk remote sensing, this study employs UAV-derived orthophotos for mining area extraction. Central to the proposed approach is the novel Gather–Injection–Perception (GIP) module, designed to overcome the information loss typically associated with conventional feature pyramid modules during feature fusion. The GIP module effectively enriches semantic features, addressing a crucial information limitation in existing methodologies. Furthermore, the network introduces the Boundary Perception (BP) module, uniquely tailored to tackle the challenges of blurred boundaries and imprecise localization in mining areas. This module capitalizes on attention mechanisms to accentuate critical high-frequency boundary details in the feature map and synergistically utilizes both high- and low-dimensional feature map data for deep supervised learning. The suggested method demonstrates its superiority in a series of comparative experiments on a specially assembled dataset of research area images. The results are compelling, with the proposed approach achieving 90.67% precision, 92.00% recall, 91.33% F1-score, and 84.04% IoU. These figures not only underscore the effectiveness of suggested model in enhancing the extraction of open-pit granite mining areas but also provides a new idea for the subsequent application of UAV data in the mining scene.

**Keywords:** open-pit granite mine; mining; quarry; unmanned aerial vehicles; multi-scale feature fusion; attention mechanisms

## 1. Introduction

Open-pit mining plays a pivotal role in supplying raw materials for various sectors including construction, municipal engineering, and industrial production. However, this form of mining poses potential threats to the ecological balance, impacting soil, water, and air quality [1–3]. Monitoring and assessing changes in open-pit mines are therefore imperative. This involves the crucial task of tracking the extent of mining operations. Such monitoring helps in identifying instances of excessive mining that adversely affect the local environment [4–6]. Furthermore, integrating these data with the 3D point cloud information of the mine enables accurate volume calculations [7].

In open-pit mining, different types of rocks exhibit distinct characteristics in remote sensing image data. Image information formed by the red (R), green (G), and blue (B) bands can provide rich feature information. Techniques such as Principal Component Analysis (PCA), band ratio, and false-color synthesis can be employed, even in multispectral remote sensing imagery, to enhance the discrimination of different rock units [8]. Traditionally,

data collection in these areas involves using total stations or Global Navigation Satellite Systems (GNSS-RTK) to create three-dimensional models, yielding detailed images of the mining zones. However, the rugged terrain of open-pit mines often makes data acquisition in steep areas challenging. These mining environments pose safety risks and are associated with high costs and time-consuming data collection processes. Furthermore, such methods yield limited data points, insufficient for a comprehensive characterization of the area [9]. Consequently, remote sensing, a non-intrusive method that does not physically interact with the data source or disrupt mining activities, has emerged as the preferred approach for data acquisition in mining. Researchers have carried out long-term exploration to solve the feature extraction problem in remote sensing images. For example, Chen [10] employed various supervised classification techniques such as maximum likelihood estimation (MLE), minimum distance classification (MDC), and support vector machine (SVM) on GaoFen-1 satellite images. This study examined the differences in principles, technologies, processes, and accuracy of these methods. Maxwell et al. [11] explored combining SVM with random forests for land cover classification in mountaintop open-pit coal mines, utilizing NAIP orthophotos and RapidEye satellite images. They discovered that SVM could effectively complement random forests in classifying land cover of surface mines. Cheng [12] utilized GaoFen-1 satellite imagery and SVM classification to analyze land cover in open-pit mining areas and to assess ecological restoration. Recent advances in deep learning have brought convolutional neural networks (CNNs) into the spotlight. Chen et al. [13], for instance, developed an enhanced UNet+ network structure and conducted experiments on GaoFen-2 images to improve information mapping accuracy in complex open-pit mining environments. Xie et al. [14] used GaoFen-2 satellite images to create a semantic segmentation dataset for open-pit mines through manual annotation, proposing a UNet-based pixel-level semantic segmentation model. Ren et al. [4] introduced a model based on an expectation maximization attention network and a fully connected conditional random field. Xie et al. [15] presented DUSegNet, a new network for segmenting open-pit mining areas, which synergizes the strengths of SegNet, UNet, and D-LinkNet, showing competitive performance on GaoFen-2 images. Liu et al. [16] proposed an integrated framework for small object detection and drivable area segmentation in open-pit mining, incorporating a lightweight attention module to enhance focus on small objects' spatial and channel dimensions, without slowing down inference. Li et al. [17] developed a siamese multi-scale change detection network (SMCDNet) with an encoder–decoder architecture for change detection in open-pit mines, emphasizing the integration of low-level and high-level change features. Satellite images have a long interval between monitoring the same area. Furthermore, the extensive nature of them complicates the delineation of boundaries in mining areas. In contrast, Unmanned Aerial Vehicles offer close-range, multi-perspective, and time-efficient imaging for terrain analysis and mineral exploration [18,19]. Eppelbaum et al. [20] utilized UAV magnetic field and Very Low Frequency (VLF) detection technology to obtain unique geological geophysical information. They proposed a new complex environmental interpretation system for locating targets in noisy backgrounds and eliminating the influence of terrain undulations, used to search for useful minerals. Thiruchittampalam et al. [21] used UAV remote sensing technology to characterize coal mine waste, extracting texture and spectral features from real-time on-site data, and employing machine learning algorithms combined with expert experience for waste classification. Kou et al. [22] used high-resolution images obtained by UAVs to identify acidic mine drainage in coal mining areas, comparing three methods—SVM, Random Forest (RF), and UNet—and proposing an efficient and economical monitoring approach. Utilizing oblique photography, these UAVs can create 3D models and generate point cloud data [7,23]. This process yields an abundance of high-quality images and digital products [24]. Deep learning techniques are instrumental in pinpointing mining areas and streamlining the computation in 3D point clouds. Moreover, the model construction process, requiring numerous photos, lays the groundwork for deep learning datasets [23,25,26]. Thus, the integration of deep learning in open-pit mining research and application is highly valuable.

This study takes the No.2 mine located in Xiling, Huashan Township, Zhongshan County, Hezhou City, Guangxi Province, as the study area, which is an open-pit granite mine. A drone is employed to collect and establish a dataset. UAV imagery, offering centimeter-level resolution compared to the meter-level resolution of remote sensing images, enhances the precision in identifying mining areas. This high accuracy expands the potential for applications across various fields. Nonetheless, the detailed nature of high-resolution images includes a wide array of land features, like water bodies, soil, vegetation, and other elements, which pose challenges to the extraction process. And the research conducted in this study has revealed that the use of feature pyramid networks (FPN) or similar structures in high-resolution UAV imagery for multi-scale feature fusion presents a challenge of information loss. FPN, which downsamples the image at multiple levels and combines multi-scale feature maps, reduces its size but at the same time sacrifices some of the original data [27–29]. When it comes to information exchange between different levels, obtaining data from non-adjacent levels requires an indirect path through intermediate layers. This indirect transmission hinders the impact of information from feature maps beyond adjacent levels on the current level. Moreover, the recursive transmission of information further aggravates information loss, resulting in the suboptimal utilization of data provided by features at different scales and an inadequate identification of mining areas. Moreover, in the high-resolution low-altitude drone imagery of mining areas, there are different interfering objects, such as mining trucks, minerals, water bodies, and bare soil. It is crucial to distinguish these objects from the mining area itself. This is in contrast to remote sensing images of satellites, where the features of the mining area appear as a unified whole. Therefore, there is a need for higher standards in accurately positioning the mining area and identifying its boundaries.

Based on the above research and analysis, this study introduces the Gather–Injection– Perception Net (GIPNet) to overcome challenges in extracting information from open-pit granite mining areas using UAV aerial images. GIPNet consists of three stages: the feature-extraction, -processing, and -decoding stages. The feature-extraction stage leverages the corresponding deep learning backbone. In the feature-processing stage, the GIP module is crafted to handle both low-dimensional local features and high-dimensional semantic features, thereby preserving more information during multi-scale feature fusion. In the feature-decoding stage, spatial upsampling attention and the introduction of boundary loss for boundary aggregation supervision are incorporated to enhance the model's ability to recognize boundaries. Enhancements are introduced to optimize information retention within the GIP module. The conventional pooling downsampling operations are replaced by a discrete wavelet transform module, preserving more information in the channel dimension while reducing image size. A dual-branch attention module segregates features into high-frequency local information and low-frequency global information, injecting them into different levels of feature maps as complementary details. The overall GIP module is further refined with the addition of a perception stage, where results from the fusion of multiple feature maps directly engage in information interaction, complementing new features generated in the injection stage. This refinement enables the model to more effectively address information loss during multi-scale feature fusion. Transitioning to the feature-decoding stage, the BP module is designed with two branches corresponding to two loss functions to better distinguish boundaries in open-pit mining areas. The proposed upsampling attention branch activates significant areas in the image using the sigmoid function during the overlay process of feature maps at different scales. The boundary aggregation branch enhances the spatial boundary details of advanced semantic features through attention mechanisms, enriching the semantic content of lower-level features. The outputs of both branches contribute to model training. GIPNet is applied to an open-pit granite mining area in Xiling, Huashan Township, Zhongshan County, Hezhou City, Guangxi Province, using a dataset of drone orthophoto maps. The study aims to enhance the extraction efficiency of mining areas by focusing on two aspects: preserving information from multi-scale feature fusion and improving the identification and positioning capabilities

of open-pit mining area boundaries. This provides a basis for mining management and ecological environment protection.

## 2. Methodology

### 2.1. Motivation

The study tackles challenges related to extracting high-resolution images from mining areas. It aims to enhance the effectiveness of mine area identification by introducing the proposed GIPNet in the research context of open-pit granite mines. The overall technical process is illustrated in Figure 1. Initially, raw image data collected by drones is imported in the data preparation phase. The orthophoto of the study area is generated through steps such as aerotriangulation and three-dimensional reconstruction. Subsequently, the pertinent open-pit mining region areas in the orthophoto are chosen for initial cropping to diminish interference from irrelevant data. The cropped images are labeled and split into training, validation, and test datasets. They are preprocessed to acquire the labeled datasets. In the proposed GIPNet processing workflow, the training dataset is inputted to build and train the GIPNet. Further optimization is conducted on the validation dataset. Upon convergence of the training loss, the associated model weights are saved and loaded. The model's performance on the test dataset is assessed, and the corresponding prediction result images are generated.

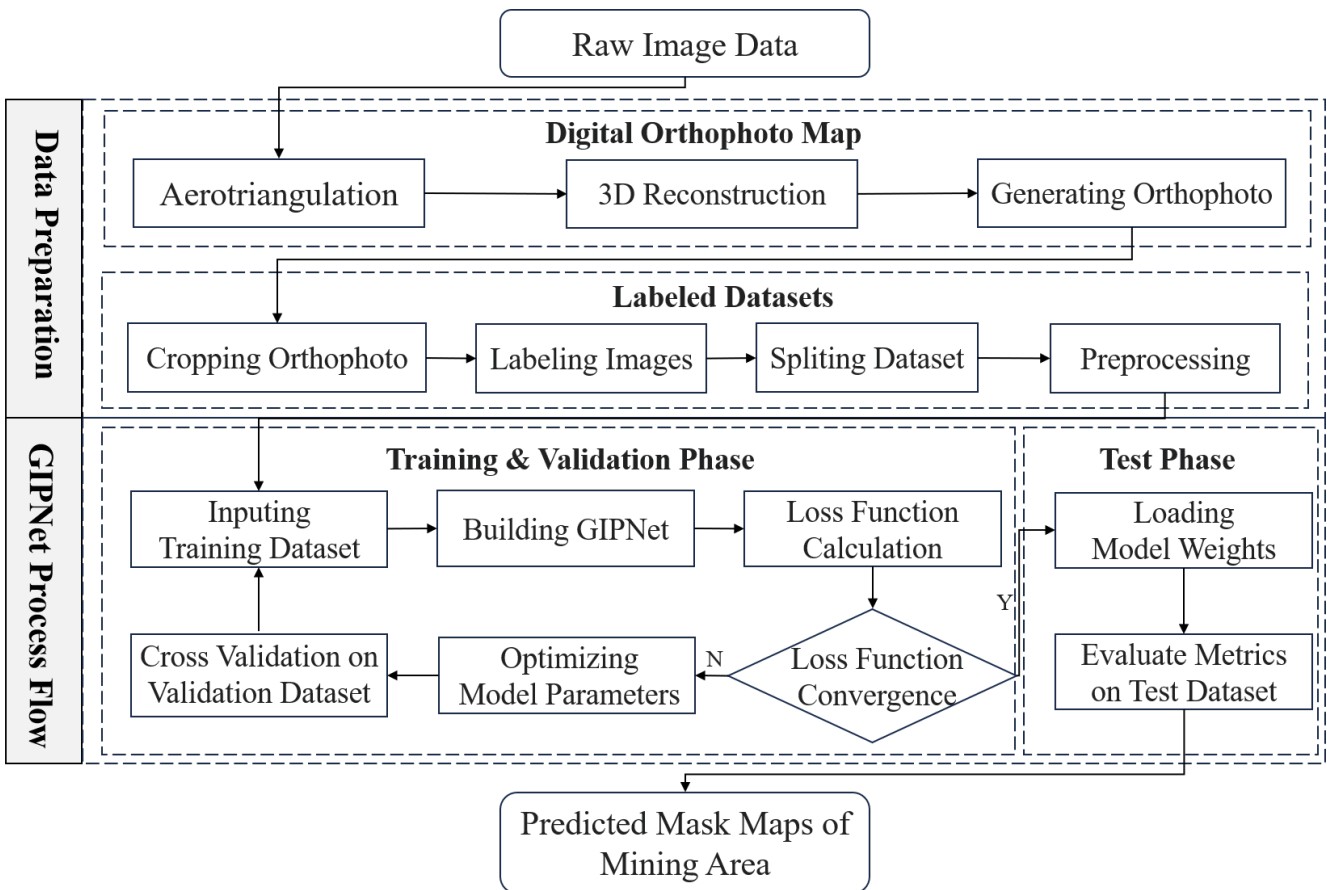

**Figure 1.** The flow diagram of the methodology.

In developing an encoder–decoder framework for segmenting open-pit granite mining areas, two critical considerations arise. The first is the integration of multi-scale features. This need stems from the substantial variation in the size and shape of mining areas across different images. In the FPN module, feature maps of various scales and channel numbers are aligned to a consistent channel number. They are then combined through

element-wise addition. This technique effectively transmits information between adjacent levels. Yet, for information exchange across multiple levels, interactions between non-adjacent levels are limited to indirect connections via intermediate layers. Consequently, this approach reduces the impact of non-adjacent level feature maps on the current layer. Drawing inspiration from recent studies [30], the suggested methodology shifts focus to intermediate layers in the feature maps. Unlike previous methods focusing on the bottom layer, the suggested approach facilitates the merging of information from two adjacent feature maps, aiming to decrease information loss.

The second major issue pertains to the blurring of boundaries and imprecise localization of mining areas. In high-resolution, low-altitude drone imagery, the variety of terrains, landforms, and features presents a considerable challenge in accurately identifying mining areas. These factors contribute to less accurate identification results. To mitigate these challenges, the Boundary Perception module bolsters the model's ability to localize mining areas more precisely by employing attention and deep supervision mechanisms, thereby enhancing the differentiation of mining area boundaries.

Figure 2 displays the detailed structure of GIPNet. The Gather–Injection–Perception (GIP) module selectively outputs feature maps of moderate sizes, enabling the integration of additional adjacent feature maps for information fusion. This module also includes a Low Stage Branch and a High Stage Branch to improve the object recognition of various sizes within the image, each incorporating gather, injection, and perception processes. Feature maps from the GIP module are then fed into the Boundary Perception (BA) module. Here, the Upsampling Attention (UA) mechanism up-samples the multi-scale feature maps while utilizing attention to highlight salient regions. Meanwhile, the Boundary Aggregation (BA) component combines features from both lower and higher levels. This integration compensates for the lack of spatial boundary information in high-level features and enhances the precision in identifying mining area boundaries.

### 2.2. Gather–Injection–Perception Module

The GIP module consists of three key processes: gather, injection, and perception. In the gather stage, the goal is to collect comprehensive information from both lower and higher levels. This involves aligning multiple input feature maps to the same scale and concatenating them along the channel dimension in both Low Gather and High Gather steps. The subsequent Low Fuse and High Fuse stages merge images from various channels to create the global information. The injection process enhances the traditional interaction in FPN by fusing global information with feature maps at different levels. The perception stage focuses on adding extra global information to offset loss incurred during indirect propagation. The inputs include feature maps $F_2$, $F_3$, $F_4$, $F_5$, extracted from the five-stage backbone network, where each $F_i$ belongs to $R^{B \times C_{F_i} \times \mathcal{R}_{F_i}}$. Here, $B$ represents the batch size, $C$ signifies the channels, and $R$, denoting dimensions, is calculated as $H \times W$. The dimensions of $R_{F_1}$, $R_{F_2}$, $R_{F_3}$, $R_{F_4}$, and $R_{F_5}$ are $\frac{R}{2}$, $\frac{R}{4}$, $\frac{R}{8}$, $\frac{R}{16}$, and $\frac{R}{32}$, respectively.

#### 2.2.1. Low Stage Branch

Based on the configuration of input feature maps in [30], this branch only uses $F_2$, $F_3$, $F_4$, and $F_5$ generated by the backbone as inputs to gather detailed information about the target object at lower levels, as shown in Figure 3.

**Low Gather Module**. This module employs intermediate feature map sizes of $F_4$ and $F_3$ to generate global information $LG_4$ and $LG_3$, respectively, as illustrated in Figure 3a,b. A significant difference from Gold-YOLO [30] is the use of the Discrete Wavelet Transform (DWT) for processing downscaled feature maps larger than $R_{F_4}$ and $R_{F_3}$. Traditional downsampling through pooling can result in the loss of high-frequency information [31]. In contrast, wavelet transform, a mathematical method for signal decomposition, separates signals into various frequency components represented by wavelet coefficients in images [32,33]. An example is Haar filtering, which performs convolution-like operations using four filters: a low-pass filter $f_{LL}$ and three high-pass filters $f_{LH}$, $f_{HL}$, and $f_{HH}$. No-

tably, these filters utilize a stride of 2 for downsampling. The Haar filter definition is detailed further below.

$$f_{LL} = \begin{bmatrix} 1 & 1 \\ 1 & 1 \end{bmatrix} \quad f_{LH} = \begin{bmatrix} -1 & -1 \\ 1 & 1 \end{bmatrix} \quad f_{HL} = \begin{bmatrix} -1 & 1 \\ -1 & 1 \end{bmatrix} \quad f_{HH} = \begin{bmatrix} 1 & -1 \\ -1 & 1 \end{bmatrix} \tag{1}$$

These four filters are orthogonal and create a reversible matrix. The Discrete Wavelet Transform (DWT) functions in the following manner: $x_{LL} = (f_{LL} \otimes x)_{\downarrow 2}$, $x_{LH} = (f_{LH} \otimes x)_{\downarrow 2}$, $x_{HL} = (f_{HL} \otimes x)_{\downarrow 2}$, $x_{HH} = (f_{HH} \otimes x)_{\downarrow 2}$.

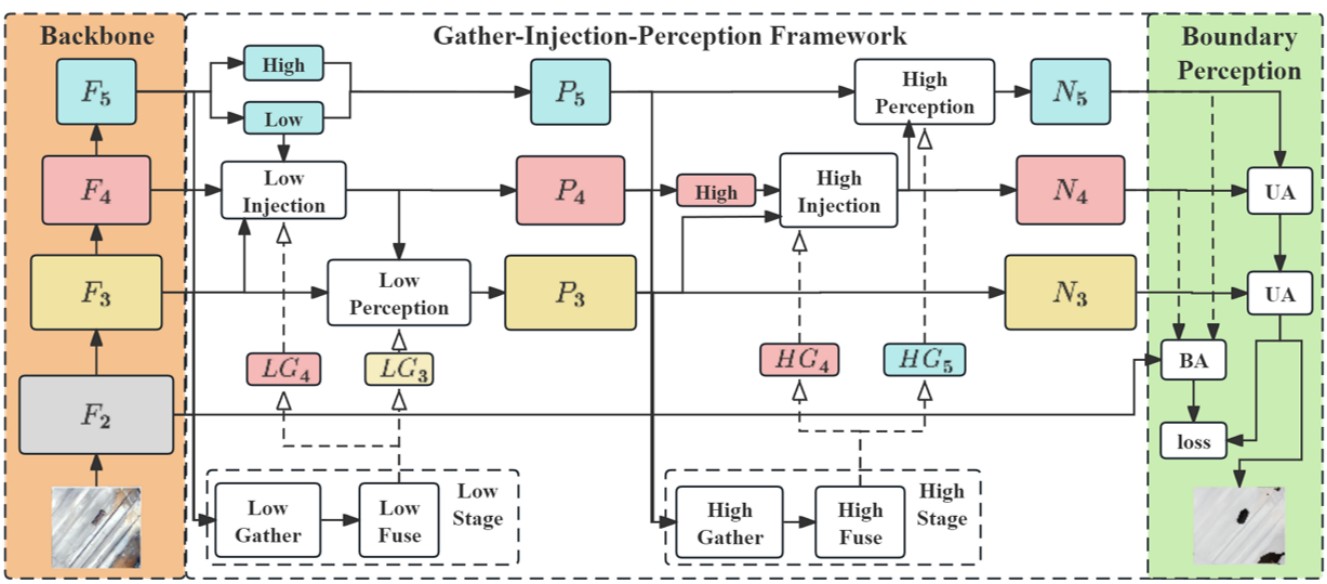

**Figure 2.** The architecture overview of proposed Gather–Injection–Perception Net(GIPNet) for open-pit granite mining area extraction based on UAV aerial images. $F_i$, $P_i$, and $N_i$ are symbols representing feature maps at different stages, where $i$ indicates the scale of the feature map in the backbone. When an image is input into the network, different scale feature maps $F_i$ are obtained through the backbone section. Gray, yellow, red, and blue colors represent feature maps at different levels, and the size of the rectangles, along with $i$, distinguishes the scales. These feature maps are input into the GIP framework for multi-scale fusion. $LG_i$ and $HG_i$ represent global information at low stage and high stage. First is the low stage, where $F_2$ to $F_4$ generate global information $LG_i$ at two scales, and colors indicate the subsequent paths they follow. **High** and **Low** in the figure represent the division of feature maps into high-frequency and low-frequency information. Dashed lines denote the operation of injecting global information $LG_i$ and $HG_i$ into the feature maps. After the low stage, the obtained feature maps are $P_3$, $P_4$, and $P_5$, which serve as inputs to the high stage, ultimately yielding $N_3$, $N_4$, and $N_5$. Finally, in the Boundary Perception stage for decoding features, $F_2$, $N_4$, and $N_5$ are inputs for Boundary Aggregation (BA), while $N_3$, $N_4$, $N_5$ are inputs for Upsampling Attention (UA). The results from both modules contribute to model training. The prediction results are output from the UA module.

Here, $x$ symbolizes the input two-dimensional image matrix. The symbol $\otimes$ represents the convolution operation, and $\downarrow 2$ signifies standard downsampling by a factor of 2. Essentially, the DWT involves four predetermined convolution filters, each with a stride of 2, to execute the downsampling process. As per Haar transform theory, the values of $x_{LL}$, $x_{LH}$, $x_{HL}$, and $x_{HH}$ at a given position (i, j) after undergoing a two-dimensional Haar transform are defined by the subsequent formulas.

$$\begin{cases} x_{LL}(i,j) = x(2i-1,2j-1) + x(2i-1,2j) + x(2i,2j-1) + x(2i,2j) \\ x_{LH}(i,j) = -x(2i-1,2j-1) - x(2i-1,2j) + x(2i,2j-1) + x(2i,2j) \\ x_{HL}(i,j) = -x(2i-1,2j-1) + x(2i-1,2j) - x(2i,2j-1) + x(2i,2j) \\ x_{HH}(i,j) = x(2i-1,2j-1) - x(2i-1,2j) - x(2i,2j-1) + x(2i,2j) \end{cases} \quad (2)$$

The terms $x_{LL}$, $x_{LH}$, $x_{HL}$, and $x_{HH}$ correspond to four downsampled images. They retain various frequency information: $x_{LL}$ for low-frequency details in both horizontal and vertical directions; $x_{HL}$ for high-frequency in horizontal and low-frequency in vertical; $x_{LH}$ for low-frequency in horizontal and high-frequency in vertical; and $x_{HH}$ for high-frequency information in both directions. The Discrete Wavelet Transform (DWT) incorporates a downsampling phase, yet thanks to the orthogonality of its filters, it allows for the original image to be losslessly reconstructed from these components. The mathematical representation of this process is detailed below.

$$\begin{cases} x(2i-1,2j-1) = \dfrac{x_{LL}(i,j) - x_{LH}(i,j) - x_{HL}(i,j) + x_{HH}(i,j)}{4} \\ x(2i-1,2j) = \dfrac{x_{LL}(i,j) - x_{LH}(i,j) + x_{HL}(i,j) - x_{HH}(i,j)}{4} \\ x(2i,2j-1) = \dfrac{x_{LL}(i,j) + x_{LH}(i,j) - x_{HL}(i,j) - x_{HH}(i,j)}{4} \\ x(2i,2j) = \dfrac{x_{LL}(i,j) + x_{LH}(i,j) + x_{HL}(i,j) + x_{HH}(i,j)}{4} \end{cases} \quad (3)$$

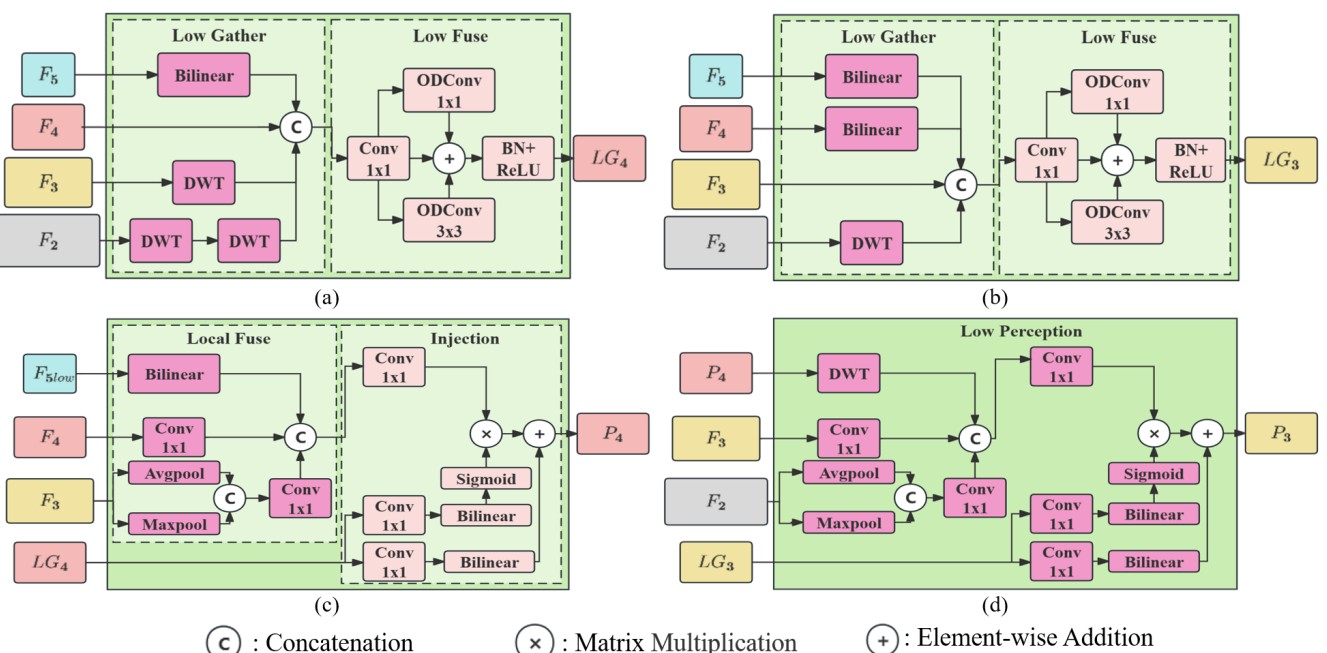

**Figure 3.** The structure of GIP Low Stage Branch Modules. (**a**) Low Gather Module base on $F_4$; (**b**) Low Gather Module base on $F_3$; (**c**) Low Injection Module; (**d**) Low Perception Module. DWT represents Discrete Wavelet Transform. $LG_i$ means global information at low stage with a size of $R_{F_i}$. $F_{5LL}$, also represented in (**c**) as $F_{5low}$, represents the low-frequency feature on the $F_5$ treated by DWT.

This suggests that using $x_{LL}$, $x_{LH}$, $x_{HL}$ and $x_{HH}$, one can infer the pixel values at any location in the two-dimensional image matrix $x$. As a result, applying DWT makes it feasible to adjust the dimensions of $F_2$ to align with those of $F_3$ and $F_4$, and similarly for $F_3$ with $F_4$. Nevertheless, the wavelet transform's effective information preservation incurs an

increase in channel dimensions. Following a single wavelet transform, the height and width of images are reduced by half, while the channel count quadruples from $C$ to $4C$, leading to higher computational demands. To address this, in the initial stage, $F_{2HH} \in R^{\mathcal{B} \times \mathcal{C}_{F_2} \times \frac{R_{F_2}}{2}}$, derived from the wavelet transform, is chosen for subsequent multi-level DWT iterations to match $F_4$'s size, as depicted in Figure 3a.

For aligning feature maps of smaller scales, such as adjusting $F_5$ to match $F_4$'s scale and then aligning both $F_5$ and $F_4$ with $F_3$, bilinear interpolation is utilized. The aligned feature maps are then concatenated, as depicted in Figure 3a,b.

**Low Fuse Module**. This module represents a departure from the approach used in Gold-YOLO [30]. Instead of traditional methods, $1 \times 1$ and $3 \times 3$ multidimensional attention dynamic convolutions, termed ODConv, are implemented on the RepVGG architectural foundation [34,35]. These replace the original convolutions. ODConv enables the learning of specific attentions in various dimensions: spatial, input channel, output channel, and convolution kernel quantity. ODConv is detailed in Equation (4).

$$y = \left( \alpha_{w_1} \odot \alpha_{in_1} \odot \alpha_{out_1} \odot \alpha_{s_1} \odot W_1 + \ldots + \alpha_{w_n} \odot \alpha_{in_n} \odot \alpha_{out_n} \odot \alpha_{s_n} \odot W_n \right) \times x \quad (4)$$

The Low Fuse Module encompasses several attention mechanisms on different dimensions of the convolutional kernel. Attention is allocated as follows: $\alpha_{w_i}$ for the convolution weight dimension, $\alpha_{in_i}$ for the input image channel, $\alpha_{out_i}$ for the output image channel, and $\alpha_{s_i}$ for the image's spatial dimension. Element-wise product, denoted by $\odot$, is utilized across various kernel space dimensions. And $x$ means input images matrix. Detailed discussion of attention calculation is deferred to subsequent sections. This design enhances the convolution operation's ability to extract comprehensive contextual information from multiple dimensions.

The module computes results through three distinct pathways: $1 \times 1$ and $3 \times 3$ dynamic convolution, and direct input matrix processing. Post-computation, batch normalization, element-wise addition, and ReLU activation are performed, as illustrated in Figure 3a,b. This process generates low global information, expressed as $LG_4 \in R^{\mathcal{B} \times \frac{C_{F_4}}{2} \times \mathcal{R}_{F_4}}$ based on $F_4$ and $LG_3 \in R^{\mathcal{B} \times \frac{C_{F_3}}{2} \times \mathcal{R}_{F_3}}$ based on $F_3$.

**Low Injection Module**. This module leverages low-frequency images, $F_{5LL}$, derived from DWT processing. In these images, $F_3$ and $F_4$ serve as inputs for feature information learning, as depicted in Figure 3c. The process involves downsampling of $R_{F3}$, targeting the output size of $R_{F4}$. To avoid the overuse of deep recursive layers in DWT, adaptive max pooling is incorporated. This step is followed by a channel-wise concatenation to preserve critical information during downsampling. The smaller feature map of $F_5$ is resized to align with $F_4$'s dimensions using bilinear interpolation and then concatenated along the channel dimension. A $1 \times 1$ convolution is subsequently employed to modify the output channel, producing the targeted low-level local information. The final step integrates the global information $LG_4$ with the local fusion information using a $1 \times 1$ convolution and a Sigmoid activation function, culminating in $P_4 \in R^{\mathcal{B} \times \frac{C_{F_4}}{2} \times \mathcal{R}_{F_4}}$.

**Low Perception Module**. The Low Stage features interactions among four feature maps. However, in fusing $F_3$ and $P_4$, there is an indirect and insufficient capture of information from $F_5$. This shortfall persists despite strategies to select intermediate feature maps that aim to cover adjacent levels. To overcome this, a new integration approach is needed. It involves combining $P_4$ with $LG_3$—derived from the Low Gather process based on $F_3$. This integration is part of the local information fusion, as illustrated in Figure 3d. Feature maps larger than $F_3$ undergo DWT, while smaller ones are resized using bilinear interpolation. The final step involves a $1 \times 1$ convolution to refine the output channel count, producing $P_3 \in R^{\mathcal{B} \times \frac{C_{F_3}}{2} \times \mathcal{R}_{F_3}}$.

Therefore, the general formula for the Low Stage Branch is shown below.

$$
\begin{aligned}
F_{4\_align} &= \text{Low\_Gather}([F_2, F_3, F_4, F_5]), & F_{3\_align} &= \text{Low\_Gather}([F_2, F_3, F_4, F_5]) \\
LG_4 &= \text{Low\_Fuse}(F_{4\_align}), & LG_3 &= \text{Low\_Fuse}(F_{3\_align}) \\
F_{5\_low} &= \text{DWT}(F_5), & P_5 &= \text{Concat}(F_{5\_low}, F_{5\_high}) \\
P_4 &= \text{Low\_Inj}(F_3, F_4, F_{5\_low}, LG_4), & P_3 &= \text{Low\_Perc}(F_2, F_3, P_4, LG_3)
\end{aligned}
\tag{5}
$$

In summary, the design of the proposed Low Stage Branch significantly deviates from the downsampling method utilized in Gold-YOLO [30], which predominantly employs adaptive average pooling to modify feature map sizes. This simpler pooling approach might result in the loss of critical information, inadequately addressing the challenges that the framework intends to resolve. To counter this limitation, the proposed method integrates a fully reversible discrete wavelet transform for downsampling. This technique effectively isolates low-frequency and high-frequency components, ensuring the retention of vital image details. Additionally, to rectify the issue of limited information exchange between $F_3$ and $F_4$, a perception module is incorporated. This module is specifically designed to enrich both $P_4$ and $LG_3$ with $F_5$ information, facilitating a more integrated fusion.

### 2.2.2. High Stage Branch

This branch represents a departure from the low stage, focusing more on high-dimensional semantic information in the image. It utilizes $P_3$, $P_4$, and $P_5$ as inputs, as depicted in Figure 4.

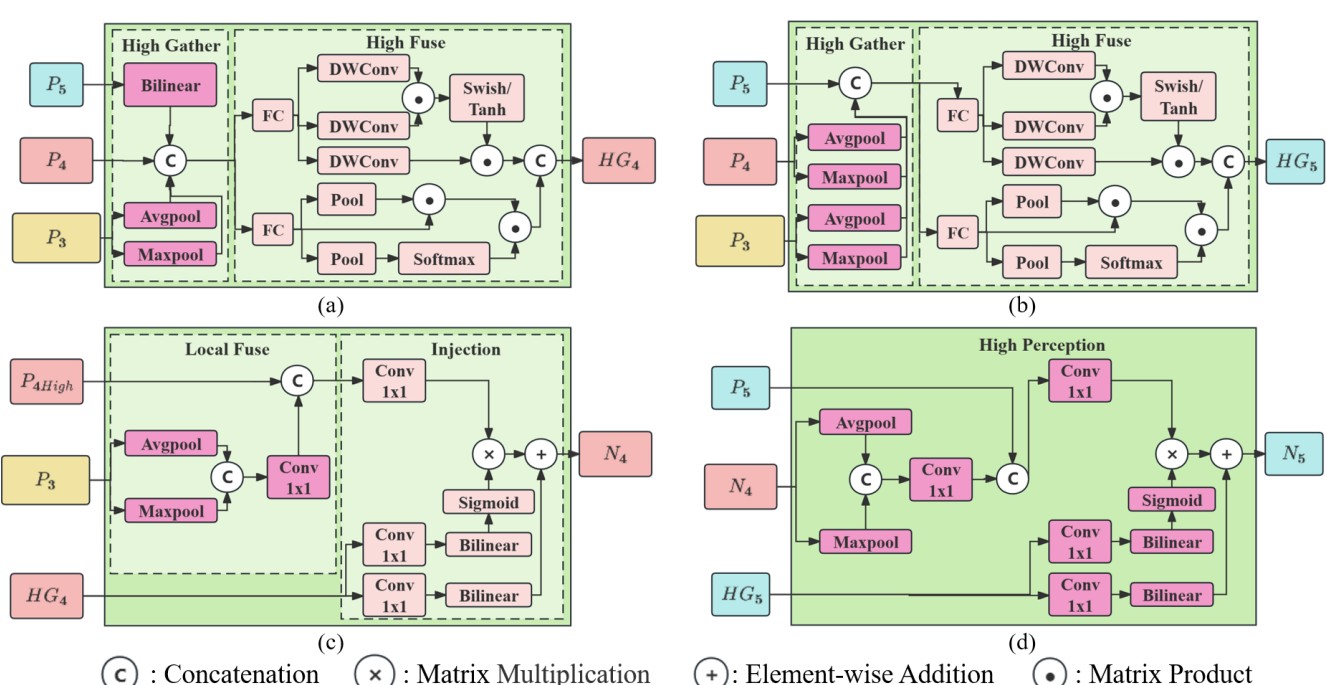

**Figure 4.** The structure of GIP High Stage Branch Modules. (**a**) High Gather Module base on $P_4$; (**b**) High Gather Module base on $P_5$; (**c**) High Injection Module; (**d**) High Perception Module. $P_i$ represents feature results from Low Stage and $N_i$ represents those from High Stage. $HG_i$ means global information at High Stage with a size of $R_{F_i}$. $P_{4High}$ represents the high-frequency feature on $P_4$ through attention operation in High Fuse Module.

**High Gather Module**. In this module, with an emphasis on higher-level information, the target outputs are set as images of $R_{P_4}$ and $R_{P_5}$. To facilitate self-attention computations and reduce computational demands, both adaptive max pooling and adaptive average pooling are applied to downscale $P_3$ and $P_4$. Adaptive max pooling is utilized to capture the

maximum value in each pooling window, highlighting prominent features and maintaining local details. Conversely, adaptive average pooling calculates the average value in each window, aiding in the preservation of overarching information while softening finer details. For resizing, small-scale feature maps are adjusted using bilinear interpolation, whereas large-scale maps are refined through the two pooling methods. Ultimately, these varied feature maps are concatenated together.

**High Fuse Module**. This module adopts a dual-branch attention mechanism to capture both high-frequency and low-frequency features from the global information at the high stage. While traditional self-attention modules are adept at capturing low-frequency global information, they struggle with high-frequency local details [36,37]. Hence, for processing low-frequency information, the standard self-attention mechanism is employed. The process begins with a linear transformation $Q, K, V = \text{FC}(P_{\text{in}})$, resulting in $Q$, $K$, and $V$ that align with conventional attention standards, where $P_{\text{in}}$ denotes the input [38,39]. In this branch, $K$ and $V$ are downscaled prior to undergoing the standard attention procedure with $Q$, $K$, and $V$. The formula is described as follows:

$$HG_{\text{low\_fre}} = \text{Attention}(Q, \text{Pool}(K), \text{Pool}(V)) \tag{6}$$

In the branch dedicated to high-frequency information, aggregation of local details initiates with DWConv processing, demonstrated by the formula $V_{\text{local}} = \text{DWConv}(V)$. Following this, $Q$ and $K$ independently gather local details, guided by the DWConv weights. The element-wise product $\odot$ is calculated between $Q$ and $K$, which then undergoes a transformation to produce context-aware weights. This phase incorporates Swish and tanh functions to add enhanced nonlinear perception capabilities. Ultimately, the synthesized weights are utilized to amplify local features, as expressed in the formula:

$$
\begin{aligned}
Q_{\text{local}} &= \text{DWConv}(Q), \\
K_{\text{local}} &= \text{DWConv}(K), \\
\text{Attn}_{\text{local}} &= \text{FC}(\text{Swish}(\text{FC}(Q_l \odot K_l))), \\
\text{Attn} &= \tanh\left(\frac{\text{Attn}_{\text{local}}}{\sqrt{d}}\right), \\
HG_{\text{high\_fre}} &= \text{Attn} \odot V_{\text{local}}
\end{aligned}
\tag{7}
$$

where $d$ represents the channel count of each token. The high-frequency local information $HG_{\text{high\_fre}}$ and the low-frequency information $HG_{\text{low\_fre}}$ are then merged to form $HG_4 \in R^{\mathcal{B} \times \frac{\mathcal{C}_{\text{F}_4}}{2} \times \mathcal{R}_{\text{F}_4}}$ and $HG_5 \in R^{\mathcal{B} \times \mathcal{C}_{\text{F}_5} \times \mathcal{R}_{\text{F}_5}}$. This process is illustrated in Figure 4a,b.

**High Injection Module**. This module serves to downsample the $P_3$ scale by utilizing both average pooling and max pooling, while also adjusting the channel dimensions. The process then merges this downscaled output with the high-frequency information $P_{4\text{high\_fre}}$, which is derived from the dual-branch attention mechanism like Equation (7). This injection procedure reflects the techniques used in the Low Stage, culminating in the integration of hierarchical and global information $HG_4$ at the F4 scale.

**High Perception Module**. This module is crafted to handle the high-dimension target size located at the edge level $F_5$. This setup results in a scenario where information transmission from $P_3$ is indirect. To manage this, a specialized perception mechanism has been integrated. The mechanism processes inputs from $N_4$ and $P_5$, subsequently enhancing the $HG_5$ based on $P_5$, ultimately leading to the creation of $N_5$.

Therefore, the general formula for the high stage branch is shown below.

$$
\begin{aligned}
P_{4\_\text{align}} &= \text{High\_Gather}([P_3, P_4, P_5]), & P_{5\_\text{align}} &= \text{High\_Gather}([P_3, P_4, P_5]) \\
HG_4 &= \text{High\_Fuse}(P_{4\_\text{align}}), & HG_5 &= \text{High\_Fuse}(P_{5\_\text{align}}) \\
N_4 &= \text{High\_Inj}(P_3, P_{4\text{high\_fre}}, LG_4), & P_3 &= \text{High\_Perc}(N_4, P_5, HG_5)
\end{aligned}
\tag{8}
$$

In conclusion, this branch exhibits advancements in attention mechanisms over the Gold-YOLO model [30]. It integrates modules that prioritize local high-frequency details, thus boosting perceptual abilities. Moreover, to tackle the challenges associated with indirect information transfer, perceptual modules have been utilized to enrich feature map information between non-adjacent hierarchical levels.

### 2.3. Boundary Perception Module

Low-level and high-level features each offer unique benefits. Low-level features, while containing less semantic detail, are rich in complex elements and are marked by clearer boundaries and reduced distortion. In contrast, high-level features are abundant in semantic information. Merging these two types of features directly can lead to redundancy and inconsistency. To address this issue, two distinct branches are developed. The first, named Upsampling Attention (UA), applies an attention mechanism to adaptively fuse features during upsampling, enhancing the richness of information at each level. The second branch, Boundary Aggregation (BA), selectively combines high-dimensional and low-dimensional features to aid in computing the boundary loss function [40]. This improves the model's ability to perceive boundaries in open-pit mining areas. The overall structure is depicted in Figure 5.

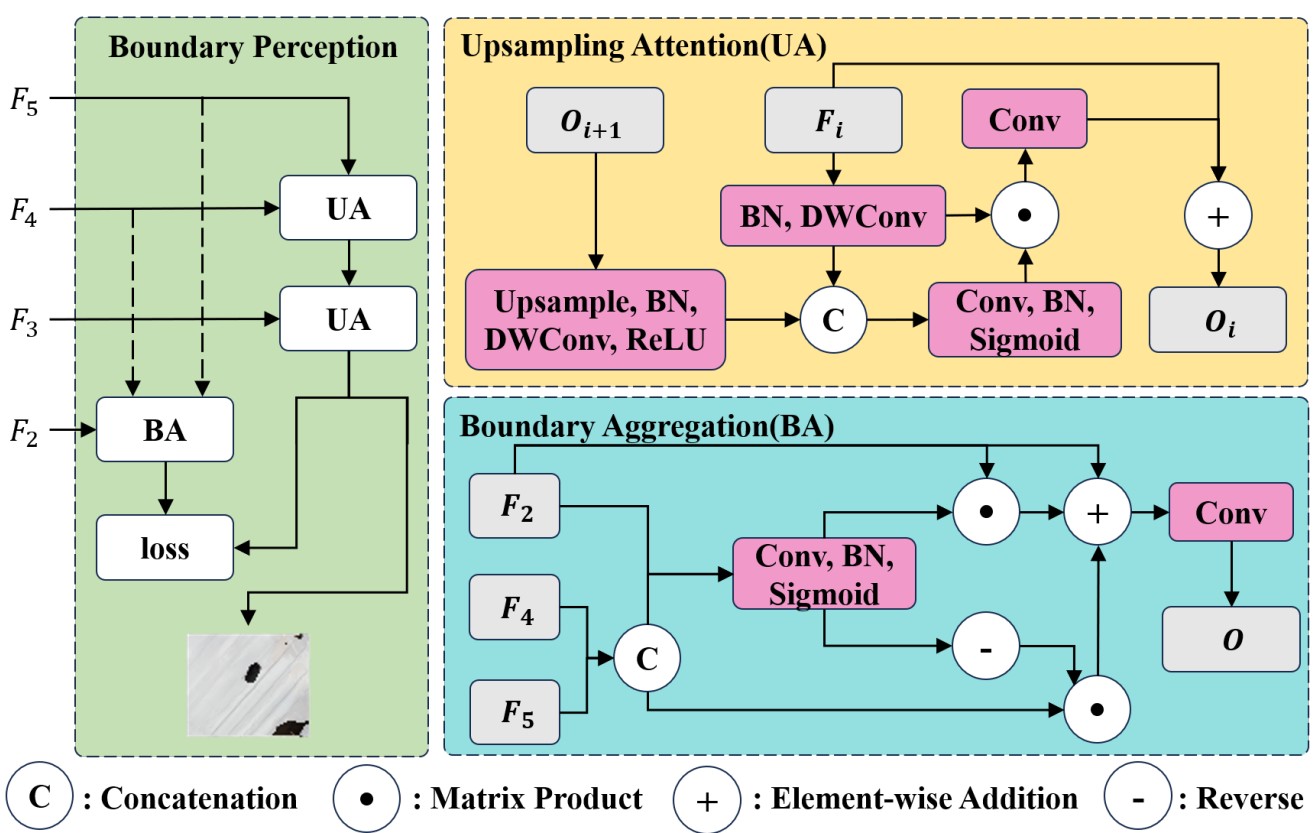

**Figure 5.** The structure of Boundary Perception Module. The feature map input to the Boundary Perception Module is denoted as $F_i$. The symbolization of $i$ is consistent with that of Figure 2. The Boundary Perception Module comprises two branches. One is **Upsampling Attention (UA)** with inputs $F_3$, $F_4$, $F_5$. In the yellow area of the figure, $O_{i+1}$ denotes the output from the previous UA module, also serving as the input for the current UA module. $O_{i+1}$ undergoes upsampling, batch normalization, DWConv convolution, and ReLU activation to produce a result. Subsequently, it is concatenated with $F_i$ after size adjustment through batch normalization and DWConv operations. Post spatial attention activation, it is element-wise added to $F_i$ to yield the output result $O_i$ of the current UA module. The other branch, **Boundary Aggregation (BA)**, takes $F_2$, $F_4$, and $F_5$ as

inputs. To distinguish the flow of $F_4$ and $F_5$ in both branches, dashed lines represent their inputs in Boundary Perception. Within the blue area depicting the BA structure, $F_4$ and $F_5$ undergo size adjustment followed by concatenation. This result is then subject to spatial attention activation with $F_2$. The output result undergoes reverse, matrix multiplication, and element-wise addition, ultimately resulting in $O$ through convolution. The outcomes from UA and BA contribute to the model's loss function computation. For generating the predicted result image, only the output from UA is required.

### 2.3.1. Upsampling Attention Branch

The UA module operates by taking the small-sized feature map $O_{i+1}$ from the high-level output and the feature map $F_i$ from the current level as inputs. It up-samples $F_{i+1}$, where $i$ is a member of the set 3, 4. Following this, $F_i$ is concatenated along the channel dimension with the upsampled map. The salient regions in the image are then activated using a Sigmoid function. The enhancement of $F_i$'s feature information is achieved by performing matrix multiplication with $F_i$, obtaining corresponding weights, and implementing element-wise addition. The related formula is presented below.

$$\begin{aligned} O_{i+1}^{\text{conv}} &= \text{DWConv}(F_i), \\ O_{i+1}^{\text{up}} &= \text{Sigmoid}(\text{BN}(\text{Concat}(\text{Upsample}(O_{i+1}), O_{i+1}^{\text{conv}}))), \\ \text{UA}(O_{i+1}^{\text{up}}, O_{i+1}^{\text{conv}}) &= \text{Conv}(O_{i+1}^{\text{up}} \odot O_{i+1}^{\text{conv}}) + F_i, \end{aligned} \tag{9}$$

Upsampling attention involves a sequence of steps beginning with the activation of a spatial attention module. This is succeeded by convolution and batch normalization. The sequence culminates with the application of the Sigmoid function. This series of steps is in harmony with the attention mechanism operating across four dimensions in the ODConv module, which is an integral part of the GIP Module.

### 2.3.2. Boundary Aggregation Branch

In the BA module, two high-level feature maps, $F_4$ and $F_5$, are chosen from four available maps. These maps are first scaled to align them, and then a channel-wise concatenation is conducted to form $F_{\text{high\_in}}$. Next, the module selects $F_2$, the feature map from the lowest level, as $F_{\text{low\_in}}$. Figure 5 illustrates how low-level and high-level information are processed separately, each contributing to the spatial saliency activation process. This branch is crucial for enhancing the spatial boundary details in high-level semantic features and enriching the semantic content in low-level features. The formula associated with this process is presented below.

$$\begin{aligned} F_{\text{low}} &= \text{Conv}(F_2), \\ F_{\text{high}} &= \text{Conv}(\text{Concat}(F_4, F_5)), \\ \text{BA}(F_{\text{low}}, F_{\text{high}}) &= F_{\text{low}} \odot F_2 + F_{\text{high}} \odot \text{Conv}(\text{Concat}(F_4, F_5)) \odot (\text{Conv}(\ominus(F_{\text{low}}))) + F_2 \end{aligned} \tag{10}$$

### *2.4. Loss Function*

Research has indicated that employing multiple loss functions with adaptive weights at different levels can significantly enhance network performance and expedite convergence [41,42]. In view of this, the suggested approach incorporates the use of cross-entropy loss and boundary loss for supervision [40]. To address challenges in boundary recognition, a weighted binary cross-entropy loss was opted for instead of the conventional Dice Loss [43]. Boundaries are determined based on pixel values, where the current value is 1, while one side is assigned 0 and the other side is assigned 1. This choice of loss function helps rectify the imbalance in boundary aggregation, reducing interference from an excessive number of 0 pixel values. The loss function is depicted in (11). In this equation, $O_{\text{UA}}$ represents the output from the UA module, while $O_{\text{BA}}$, which contributes to the boundary loss, represents the output from the BA module. The ground truth is denoted as GT. In the cross-entropy loss formula $L_{CE}(O_{\text{UA}}, GT)$, $N$ denotes the sample size, $y_i$ corresponds to the

ground truth labels, $\hat{y}_i$ denotes the model's predictions, and $\sigma$ refers to the Sigmoid function. In the boundary aggregation loss formula $L_{BD}(O_{BA}, GT)$, $w_i$ represents the weight term, which is adjusted based on the count of positive and negative samples. In the computation of $w_i$, $\text{num}_{\text{pos}}$ denotes the total count of pixels with a value of 1 in the image, $\text{num}_{\text{neg}}$ represents the total count of pixels with a value of 0, and sum indicates the total pixel count. The weight at the corresponding position is denoted as $w_i$. The weighting coefficients are indicated by $\lambda_1$ and $\lambda_2$.

$$L_{CE}(O_{\text{UA}}, GT) = -\frac{1}{N}\sum_{i=1}^{N}[y_i \cdot \log(\sigma(\hat{y}_i)) + (1-y_i) \cdot \log(1-\sigma(\hat{y}_i))],$$

$$w_i = \begin{cases} \frac{\text{num}_{\text{neg}}}{\text{sum}}, & \text{if value}_i = 0 \\ \frac{\text{num}_{\text{pos}}}{\text{sum}}, & \text{if value}_i = 1 \end{cases}'$$

$$L_{BD}(O_{\text{BA}}, GT) = -\frac{1}{N}\sum_{i=1}^{N}[y_i \cdot \log(\sigma(\hat{y}_i)) \cdot w_i + (1 - y_i) \cdot \log(1 - \sigma(\hat{y}_i)) \cdot (1 - w_i)],$$

$$L_{\text{total}} = \lambda_1 L_{CE}(O_{\text{UA}}, GT) + \lambda_2 L_{BD}(O_{\text{BA}}, GT)$$

(11)

## 3. Experiments

### 3.1. Raw Data

This research investigated No.2 open-pit granite mine located in Xiling, Huashan Township, Zhongshan County, Hezhou City, Guangxi, China, as shown in Figure 6a. The study area is located within the coordinates of 24°32′30″N to 24°34′30″N and 111°07′30″E to 111°09′30″E. The main variety of granite is Zhongshan Qing, also known as "Golden Spot Green Granite", derives its name from Zhongshan County in Hezhou City, Guangxi Province [44]. It features a dark green color, almost approaching black, with a dense structure and hard texture. The confirmed resource reserves are approximately 21.578 million m$^3$, with a prospective mining area of about 16.75 km$^2$. The potential resource reserves are estimated to be around 1.86 billion m$^3$. The area, marked in yellow on the Google Earth image (see Figure 6b). It contains 22 typical mining sites, and Figure 6c illustrates the on-site mining scene of one of these.

In this study, the DJI M300RTK unmanned aerial vehicle, outfitted with five SONY ILCE-5100 perspective cameras, was employed. Detailed specifications of the UAV and camera can be found in Figure 7 and Table 1. Five distinct sets of images were captured in the research area in 2022, each set using different camera inclination angles. These efforts resulted in a collection of 2905 images, each with a resolution of 6000 × 4000 pixels, thoroughly covering the yellow zone shown in Figure 6b. Subsequently, Context Capture software was used to perform three-dimensional reconstruction on these image sets [24]. This procedure generated various geospatial digital products, including a three-dimensional point cloud, a Digital Surface Model (DSM), and a Digital Orthophoto Map (DOM).

**Table 1.** Parameter configuration of aerial survey.

| Camera | | UAV | |
|---|---|---|---|
| Type | ILCE-5100 | Type | M300RTK |
| Sensor | 23.5 mm | Flight time | 55 min |
| Millimeter focal length | 6.56287 mm | Maximum take-off weight | 9 kg |
| Focal length in pixel | 1675.63 pixel | Maximum load | 2.7 kg |

### 3.2. Dataset

ArcGIS software was used to analyze the orthophoto map, which consists of 82,536 columns and 112,426 rows with a pixel resolution of 0.04 m [9]. In this map, 22 open-pit mining areas were identified, as highlighted by red line annotations in Figure 8a. To focus on relevant data, rectangular vector boxes in ArcGIS were applied to crop these mining areas and adjacent terrain. The cropped TIFF data were then imported into Labelme

software for consistent annotation [45]. All images remained RGB three bands, TIFF format 8 bit depth, cropped image block size was 512 × 512 and overlap factor was 256. After processing, a collection of 2762 annotated images was produced. Recognizing the complexity of features in high-resolution open-pit mining scenes, a preliminary analysis was conducted to assess the proportional distribution of these features in the dataset, as illustrated in Figure 8b. This study focuses on the granite mining area, particularly the portion labeled as "pit" in pixel terms. In high-resolution low-altitude UAV images, machines, trucks, and large rocks resulting from excavation display clear semantic distinctions from the pit, leading to a lack of semantic consistency. Consequently, this research classifies all features, excluding the pit, as the background class. The boundary of the pit class is considered as the boundary of the mining area.

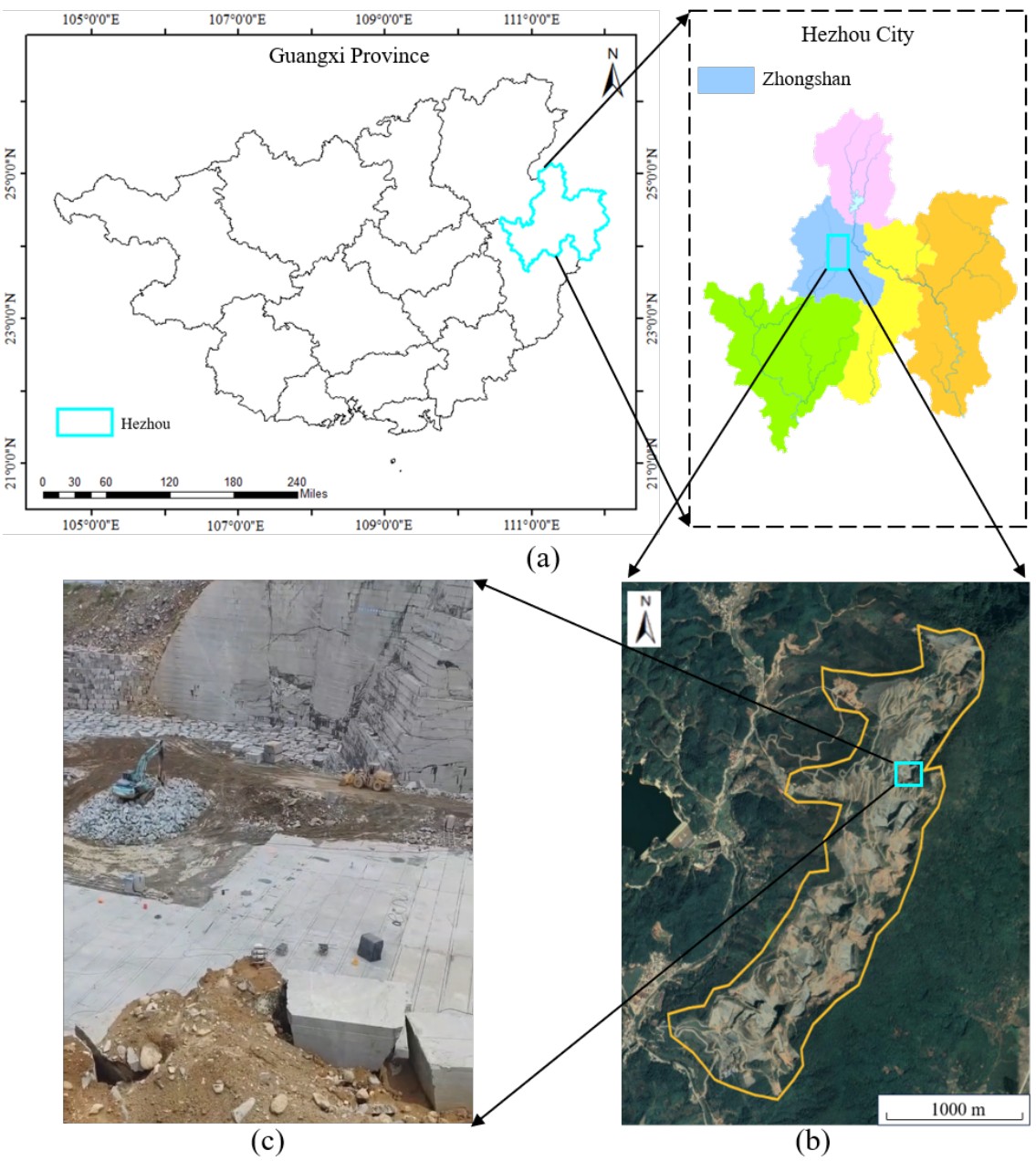

**Figure 6.** Research area and location of the open-pit granite mine. (**a**) The geographical location of the studied mining area; (**b**) an aerial view of the research mining area as viewed from Google Earth.; (**c**) a field mining scenario in the study area.

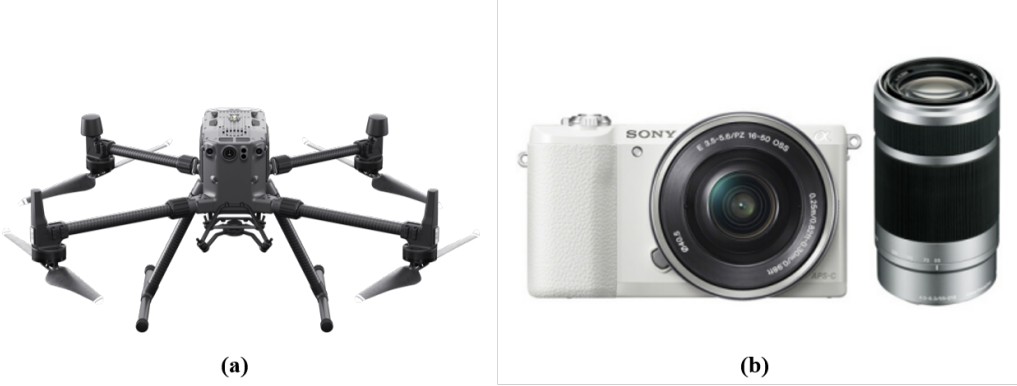

**(a)**                                                                **(b)**

**Figure 7.** Drone equipment to collect data. (**a**) DJI M300RTK. The device is manufactured by Shenzhen DJI Innovation Technology Co., Ltd., based in Shenzhen, Guangdong Province, China. (**b**) SONY ILCE-5100. The device is manufactured by SONY (China) Co., Ltd., located in Beijing, China.

### 3.3. Experiment Setup

The computing system employed in the experimental setup featured an Intel Xeon G6130H and an Nvidia GeForce RTX 3090 graphics card. This system operated on Ubuntu 20.04 and was equipped with 64 GB of RAM. Python 3.8 served as the programming language for the experimental model, while PyTorch was used as the deep learning framework.

A series of network models were selected for comparison experiments, which were shown in Table 2. Among them, the K-Net and SegNeXt models were relatively large, and the optimizer was implemented according to the original paper, using the AdamW optimizer. The encoder and decoder of UNet were composed of five layers of BasicConvBlock. The backbone of SegNeXt was a model called MSCAN proposed in the original paper, which replaced traditional convolution with convolutional attention modules. Other models not specifically mentioned used ResNet-50 as the backbone, meaning a ResNet with a depth of 50, and they utilized the Stochastic Gradient Descent(SGD) optimizer.

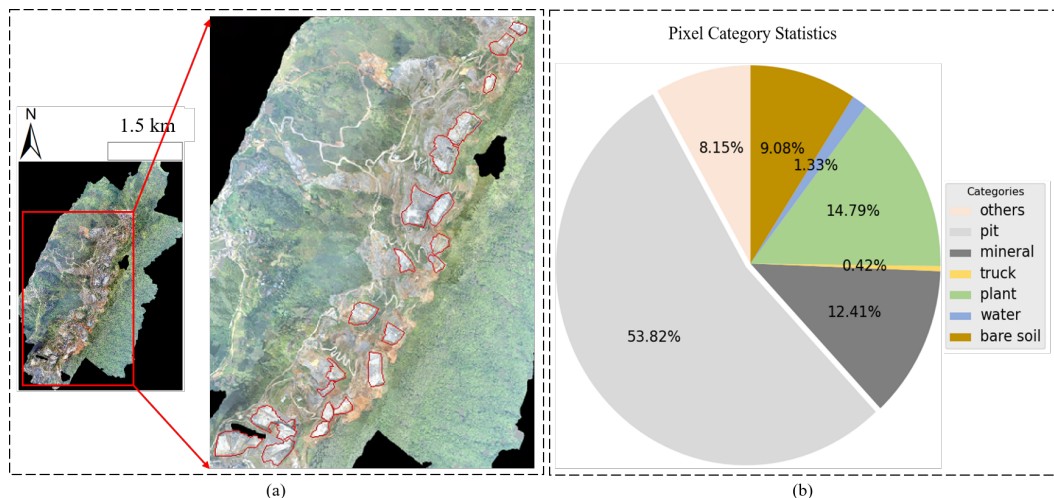

(a)                                                                (b)

**Figure 8.** Information about the study area. (**a**) 22 mining sites in the study area. They are marked in the figure using red lines. (**b**) Pixel category statistics of the labeled dataset.

The dataset was divided into three subsets: training, validation, and testing, in a 7:2:1 ratio, comprising 1933, 552, and 277 images, respectively. In the training set, data augmentation techniques such as random image flipping and photometric distortions were implemented. Each image underwent a 50% chance of random flipping. Photometric distortions involved randomly varying brightness, contrast, saturation, and hue. A consistent

batch size of 8 was used throughout the training, starting with an initial learning rate of 0.01. The momentum was set at 0.9, with a weight decay of 0.0005. The model underwent 80,000 iterations to effectively predict mining regions. During the training phase, the cross-entropy loss function and boundary loss function were used for backpropagation to optimize the GIPNet model strategy. For other models, only the cross-entropy loss function was applied.

**Table 2.** An overview of the characteristics of the network models involved in the comparison experiment.

| Network | Characteristic | Backbone | Optimizer |
|---|---|---|---|
| **K-Net** [46] | **K-Net** enhances segmentation core by dynamically updating instance kernels and mask predictions. | ResNet-50 | AdamW |
| **PointRend** [47] | **PointRend** achieves denser sampling in boundary regions and predicts point-based segmentations at adaptively chosen positions. | ResNet-50 | SGD |
| **PSPNet** [27] | **PSPNet** introduces the Pyramid Pooling Module, replacing global pooling operations and collecting information from diverse scales and sub-regions. | ResNet-50 | SGD |
| **UNet** [48] | **UNet** features a U-shaped network structure for precise localization and enhances segmentation accuracy. | 5 layers of BasicConvBlock (Conv+BN+ReLU) | SGD |
| **UPerNet** [49] | **UPerNet** enhances global prior representation by applying the Pyramid Pooling module and predict texture labels through additional convolution layers. | ResNet-50 | SGD |
| **HRNet** [50] | **HRNet** incrementally adds high to low-resolution subnetworks, connecting them to exchange information and generates rich high-resolution representations. | HRNet | SGD |
| **FCN** [51] | **FCN** replaces fully connected layers with convolutional layers for direct pixel-level predictions. | ResNet-50 | SGD |
| **DeepLabv3** [31] | **DeepLabv3** utilizes dilated convolutions to extract dense feature maps capturing long-range contextual information and introduces the Atrous Spatial Pyramid Pooling module to improve accuracy. | ResNet-50 | SGD |
| **DANet** [52] | **DANet** introduces spatial and channel attention for integration of global information to captures pixel-level spatial relationships and inter-channel correlations. | ResNet-50 | SGD |
| **SegNeXt** [53] | **SegNeXt** presents a multi-scale convolutional attention module within the conventional encoder–decoder framework, substituting the traditional self-attention mechanism. | MSCAN | AdamW |

### 3.4. Evaluation Metrics

During the experimental validation and test phase, widely used evaluation metrics in the relevant field were employed to quantitatively analyze the model's prediction results, ensuring a reliable and comprehensive assessment of the extraction of mining areas in open-pit mines [54]. In the semantic segmentation task for open-pit mining areas, the digit 0 typically represents the background region, while the digit 1 denotes the mining area. The T and P represent the ground truth and model prediction results, respectively. The pixel classification in an image can be summarized into four categories: TP (True Positive) indicates the correctly classified background pixels and their quantity, TN (True Negative) represents the correctly classified mining area pixels and their quantity, FP (False Positive) signifies the mining area pixels incorrectly classified as background along with their quantity, and FN (False Negative) denotes the background pixels wrongly classified as mining area and their quantity. The selected metrics are based on these classification

scenarios. For example, precision represents the proportion of pixels correctly classified as mining areas to the total number of pixels, as formulated below.

$$\text{Precision} = \frac{\text{TP}}{\text{TP} + \text{FP}} \tag{12}$$

Recall, which signifies the percentage of pixels predicted by the model among all mining areas in the ground truth, is formulated as follows.

$$\text{Recall} = \frac{\text{TP}}{\text{TP} + \text{FN}} \tag{13}$$

The F1-score represents the harmonic mean between Precision and Recall, formulated as follows.

$$\text{F1-score} = \frac{2 \times \text{Precision} \times \text{Recall}}{\text{Precision} + \text{Recall}} \tag{14}$$

The Intersection over Union (IoU) represents the intersection divided by the union between the model's predicted results for mining areas and the ground truth, formulated as follows.

$$\text{IoU} = \frac{\text{TP}}{\text{TP} + \text{FN} + \text{FP}} \tag{15}$$

## 4. Results

### 4.1. Comparison Experiments

To assess the GIPNet's effectiveness and rationality in segmenting open-pit granite mining areas, comparative experiments were performed using various established methods on the dataset described in Section 3. The UNet was selected as the backbone for GIPNet. Its training took 12 h.

Table 3 displays a significant improvement in evaluation metrics with the integration of the pluggable GIP framework into the backbone. Additionally, this approach surpasses the performance of other methods, showcasing superior overall results. The evaluation employs metrics such as Precision, Recall, F1-score, and IoU. GIPNet attains impressive scores: 90.67% in Precision, 92.00% in Recall, 91.33% in F1-score, and 84.04% in IoU. On F1-score and IoU, it surpasses the second position by 1.10% and 1.84%, respectively. The Precision and Recall also rank within the top two.

**Table 3.** Accuracy assessment results of comparison experiments on proposed dataset (with the bold and underlined data for the best and second-best metrics).

| Method | Precision | Recall | F1-Score | IoU |
|---|---|---|---|---|
| SegNeXt | 85.35 | 89.66 | 87.45 | 77.70 |
| UNet | 90.37 | 87.31 | 88.81 | 79.88 |
| DeepLabv3 | 86.64 | _91.56_ | 89.03 | 80.23 |
| FCN | **90.68** | 87.74 | 89.19 | 80.48 |
| UPerNet | 88.70 | 90.10 | 89.39 | 80.82 |
| DANet | 89.27 | 90.40 | 89.83 | 81.54 |
| PSPNet | 89.64 | 90.20 | 89.91 | 81.68 |
| HRNet | 89.76 | 90.49 | 90.13 | 82.03 |
| K-Net | 89.42 | 90.91 | 90.06 | 82.09 |
| Pointrend | 89.91 | 90.55 | _90.23_ | _82.20_ |
| GIPNet | _90.67_ | **92.00** | **91.33** | **84.04** |

UAV aerial images within the research area exhibit a variety of sizes and shapes in both mining areas and other objects. The performance of different deep learning models in training and prediction varies considerably. To visually showcase mining area extraction in the dataset, selected representative images are presented in Figure 9. GIPNet, the proposed model, excels in accurately delineating boundaries between mining and non-mining re-

gions, while effectively reducing misidentification and omission errors. Conversely, UNet and SegNeXt struggle with multi-scale feature fusion and perception, leading to less effective segmentation in challenging differentiation areas, as depicted in columns (c) and (d). In high-resolution open-pit mining contexts, specific interferences hinder the recognition capabilities of UNet and SegNeXt. Meanwhile, the accuracy of FCN, DeepLabv3, and PSP-Net in discerning detailed features and geometric shapes within mining areas could be enhanced, as demonstrated in columns (e), (f), (g), and (h). Although these models capture general outlines, the shapes segmented by DeepLabv3 and PSPNet lack precision due to limitations in similar feature pyramid modules. In contrast, GIPNet effectively identifies mining area contours, matches correct geometric shapes, and preserves multi-scale feature perception in complex scenes, thereby enabling efficient mining area extraction.

### 4.2. Ablation Experiments

The experimental results, both qualitative and quantitative, highlight GIPNet's exceptional ability in detecting mining areas in open-pit granite mines. To further assess the impact of the GIP Module and BA Module on the outcomes, ablation studies were performed on the dataset. These experiments aimed to validate their effectiveness. The modules were also compared with FPN [55] and Gold-YOLO [30], hereafter referred to as GD. Consequently, six experimental setups were created: (i) baseline; (ii) baseline + FPN; (iii) baseline + GD; (iv) baseline + GIP; (v) baseline + BA; (vi) baseline + GIP + BA. For consistency in comparison, all experiments used UNet as the baseline model. Figure 10 displays the visual outcomes of these ablation studies on selected images.

The absence of modules adept at multi-scale feature fusion leads to frequent false and missed segmentations. Incorporating the FPN module generally enhances recognition capabilities, especially in challenging or smaller areas. However, the use of GD sometimes shows reduced effectiveness compared to FPN. This reduction in performance is often due to the loss of detail during the feature aggregation in GD, which may create black voids in complex segmentation scenarios, as depicted in Figure 10e. The improved GIP module, on the other hand, retains more information through the multi-scale fusion process, thus enhancing recognition precision. Despite this improvement, there is room for further refinement in extracting intricate contour details, as shown in column (f). Importantly, the absence of GIP leads to persistent fragmented regions in difficult-to-distinguish areas, as column (g) illustrates. The combination of the GIP and BA modules effectively overcomes these challenges, culminating in superior recognition performance. Table 4 presents the evaluation of these modules using four distinct metrics. The improved GIP module alone leads to enhancements in both F1-score and IoU compared to the FPN and GD modules. Moreover, the integration of GIP with BA yields more substantial improvements, with enhancements of 2.52% and 4.16%, respectively, over the baseline UNet. Compared with GD, the integration of both GIP and BA leads to improvements of 1.80% and 2.99%.

Section 2 introduces that GIPNet allows the utilization of various backbones for feature extraction. ResNet, a classical convolutional neural network structure, is available in depths of 18, 34, 50, 101, and 152 [56]. Deeper networks suggest the capacity to grasp more intricate and abstract features, albeit potentially leading to overfitting and extended training durations. The recognition performance of GIPNet employing ResNet as backbones of these five depths is presented in the Table 5. Due to memory limitations, batch sizes for ResNet 101 and 152 are configured at 4, while other models with varying depths adopt a size of 8. SGD serves as the optimizer, and the models train 80,000 iterations.

The results reveal that ResNet-50 is a suitable depth for GIPNet. It outperforms the second-best by 0.73% in F1-Score and 1.19% in IoU metrics. This suggests that, within the dataset proposed in this paper, a model with a depth of 50 avoids underfitting issues resulting from inadequate training or the inability to capture deep features due to shallow depth. In contrast to ResNet-101 and ResNet-152, it prevents overfitting in training by prioritizing the effectiveness on the training set and maintaining good performance on the test set.

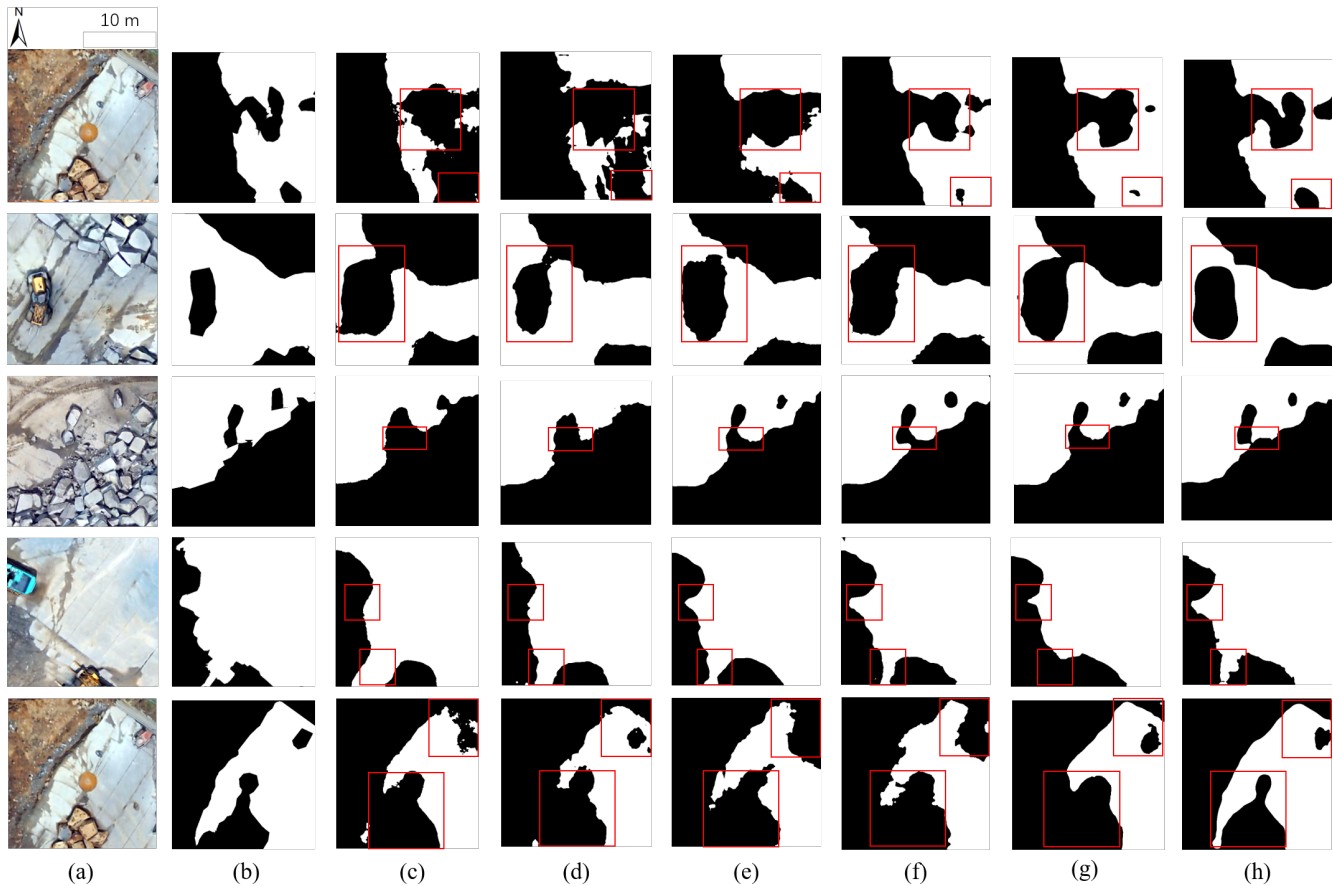

**Figure 9.** Visualization results of comparison experiment on proposed dataset. (**a**) Image; (**b**) Ground Truth; (**c**) UNet; (**d**) SegNeXt; (**e**) FCN; (**f**) DeepLabv3; (**g**) PSPNet; (**h**) GIPNet.

**Table 4.** Accuracy assessment results of ablation experiments on proposed dataset (with the bold and underlined data for the best and second-best metrics).

| Method | Precision | Recall | F1-Score | IoU |
|---|---|---|---|---|
| Baseline | 90.37 | 87.31 | 88.81 | 79.88 |
| Baseline + FPN | 89.82 | 89.28 | 89.55 | 81.08 |
| Baseline + GD | 88.62 | <u>90.46</u> | 89.53 | 81.05 |
| Baseline + GIP | **90.98** | 89.66 | <u>90.31</u> | <u>82.34</u> |
| Baseline + BA | 90.12 | 87.60 | 88.84 | 79.92 |
| Baseline + GIP + BA | <u>90.67</u> | **92.00** | **91.33** | **84.04** |

**Table 5.** Comparison of evaluation metrics results of GIPNet with different backbones on the proposed dataset (with the bold and underlined data for the best and second-best metrics).

| Backbone | Precision | Recall | F1-Score | IoU |
|---|---|---|---|---|
| ResNet-18 | **90.94** | 87.40 | 89.14 | 80.04 |
| ResNet-34 | 88.71 | 89.98 | <u>89.34</u> | <u>80.74</u> |
| ResNet-50 | <u>89.16</u> | <u>90.99</u> | **90.07** | **81.93** |
| ResNet-101 | 84.95 | **92.90** | 88.75 | 79.78 |
| ResNet-152 | 84.63 | 89.39 | 86.95 | 76.91 |

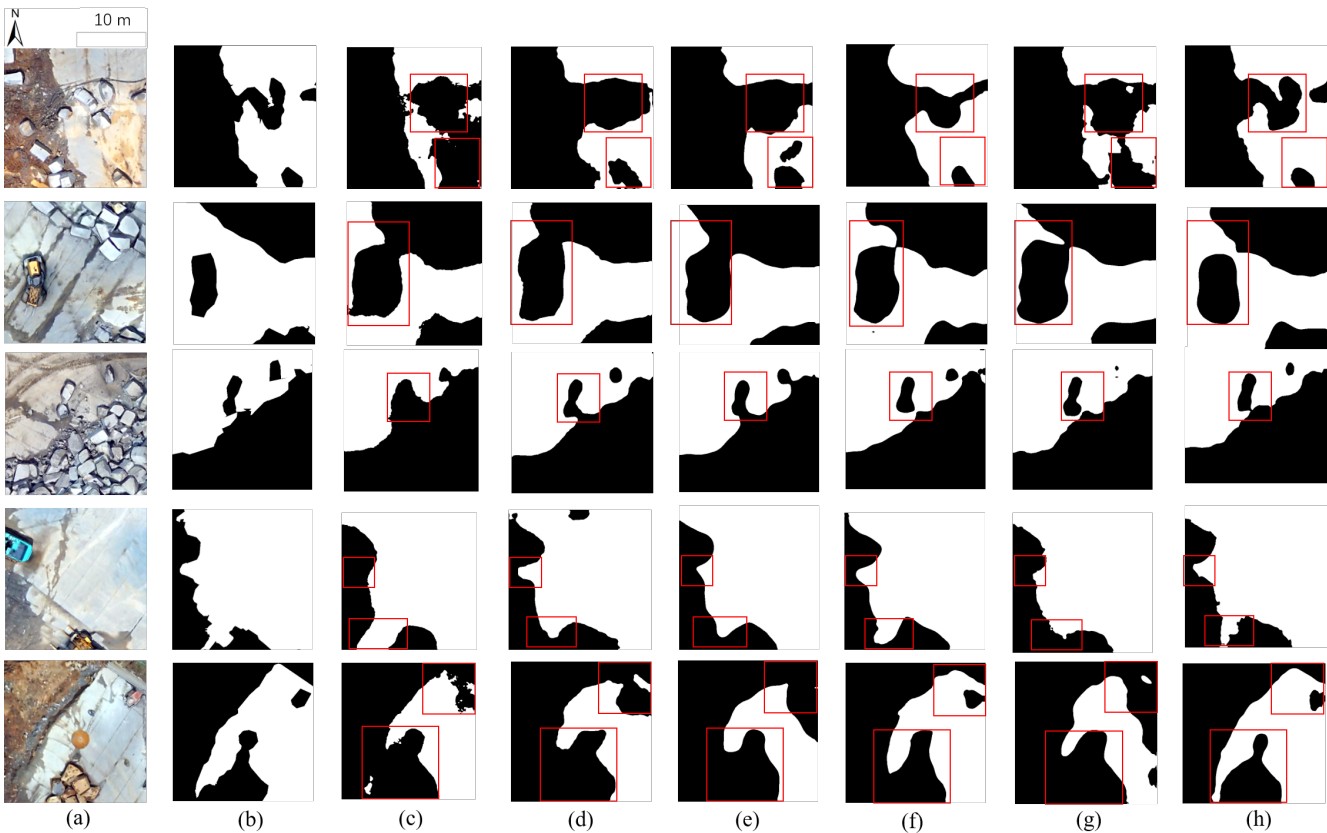

**Figure 10.** Visualization results of ablation experiments on proposed dataset. (**a**) Image; (**b**) Ground Truth; (**c**) Baseline; (**d**) Baseline + FPN; (**e**) Baseline + GD; (**f**) Baseline + GIP; (**g**) Baseline + BA; (**h**) Baseline + GIP +BA.

Moreover, considering the overarching goal of model training and optimization, the effectiveness of the cross-entropy loss and boundary loss is influenced by $\lambda_1$ and $\lambda_2$. Consequently, this study varies these parameters for comparison purposes. The relative magnitudes of them impact the model's performance in multi-scale land cover recognition and boundary localization. In this set of comparative experiments, GIPNet employs Unet as the backbone, with a batch size of 8, SGD optimizer, and 80,000 iterations, as depicted in the Table 6.

**Table 6.** Comparison of evaluation metrics results of GIPNet with different $\lambda_i$ in the loss function on the proposed dataset (with the bold and underlined data for the best and second-best metrics).

| $\lambda_1$:$\lambda_2$ | Precision | Recall | F1-Score | IoU |
|---|---|---|---|---|
| 4:1 | 88.00 | <u>92.49</u> | <u>90.19</u> | <u>82.13</u> |
| 3:2 | 83.70 | **93.48** | 88.32 | 79.08 |
| 1:1 | 87.97 | 90.66 | 89.29 | 80.66 |
| 2:3 | <u>90.33</u> | 88.62 | 89.47 | 80.94 |
| 1:4 | **90.67** | 92.00 | **91.33** | **84.04** |

Upon comparing the results, it is evident that elevating the weight ratio of boundary loss during the loss stage leads to a increase of 1.91% in the IoU over the second highest, with the highest F1-score. The GIP module has initially processed the information interaction of multi-scale fusion. Therefore, in the loss calculation of the BA module, the features obtained tend to have a marginal effect towards the high weight of $lambda_1$. If the weight is biased towards $lambda_2$, which is related to boundary recognition, it can enrich the model's recognition ability and improve the performance.

## 5. Discussion

### 5.1. Advantages and Disadvantages of Multi-Scale Feature Fusion Methods

When objects of varying sizes are subjected to the same downsampling ratio, they experience significant semantic discrepancies. This often results in less effective recognition of smaller objects. The Feature Pyramid Network (FPN) addresses this by providing different resolution levels, each tailored to represent features of objects at various sizes. This multi-scale approach substantially improves model performance. Figure 11 shows the data flow in FPN. In the downsampling phase (left side of Figure 11), the height and width decrease while the channel dimension increases, forming a hierarchical pyramid. In the decoding head, corresponding feature maps are produced during upsampling (right side of Figure 11). Each level's feature map can serve as an independent prediction output or be combined with the next level's map for multi-scale fusion. This process repeats until the final prediction is reached.

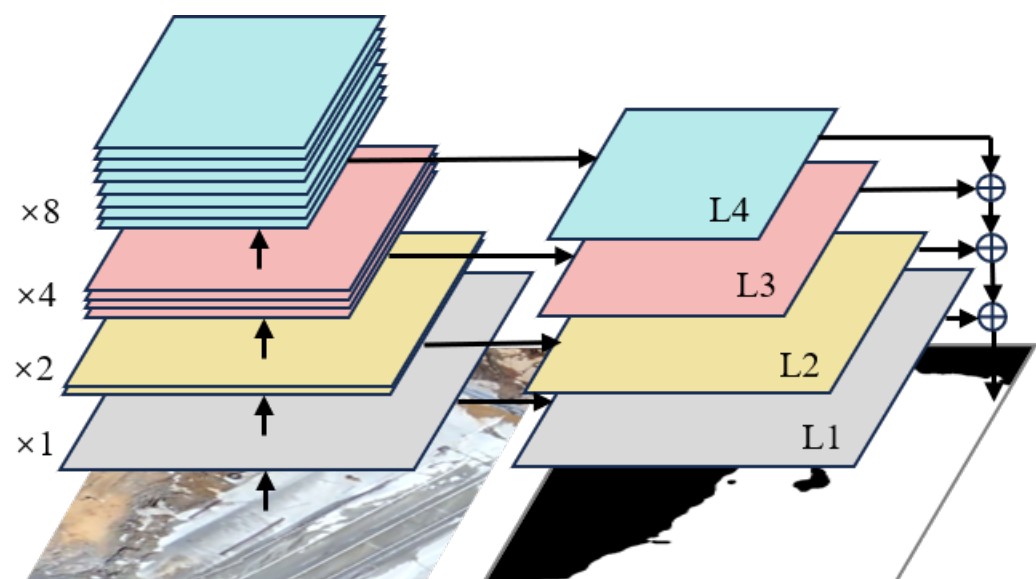

**Figure 11.** The information flow of FPN structure.

However, accessing information across multiple scales can be complex. For instance, in Figure 11, retrieving information from level L3 requires integrating features from levels L2 and L3 at level L1. Accessing level L4 involves an even longer chain of recursive calls. To address these challenges, researchers have developed several enhancements. Deeplabv3's ASPP module, for example, uses branches with different strides, followed by downsampling and channel-wise concatenation [31]. PANet introduces a bottom-up path to enrich the information flow [57]. The FPT model incorporates background objects, providing context like relative positions to assist in object classification [58]. These advancements emphasize the importance of fully integrating semantic features across levels. However, even with these improvements, FPN-based fusion structures still face limitations in cross-layer information exchange.

Inspired by the progress in TopFormer [59] and Gold-YOLO [30], this study aims to enhance information preservation. Initially, the research explored the idea proposed in [30], which advocates for the use of intermediate over edge levels in the hierarchical structuring of feature maps. This approach is crucial for accurate predictions and allows for a wider reach to neighboring layer images. Following this, it was found that modifying feature maps across different scales using basic pooling operations resulted in substantial loss of information. Specifically, methods like average and max pooling downsampling involve calculating a single value within a defined scope, replacing the original details. This process, whether using average or maximum values, tends to neglect the finer aspects of the original data. To address this, the discrete wavelet transform was adopted. This

technique decomposes an image into wavelet coefficients at varied scales and directions, capturing intricate information effectively. Furthermore, the wavelet transform skillfully segregates information into high-frequency and low-frequency elements. This segregation enhances the adaptability in applying it to the low and high branches of the GIP module. Figure 12 showcases the visual comparison of average pooling downsampling, max pooling downsampling, and discrete wavelet transform. It highlights the clear distinctions in detail preservation between average and max pooling, shown in (a) and (b), respectively. In contrast, (c) demonstrates how the discrete wavelet transform enriches the image with more high-frequency details and low-frequency contours in both the horizontal and vertical planes.

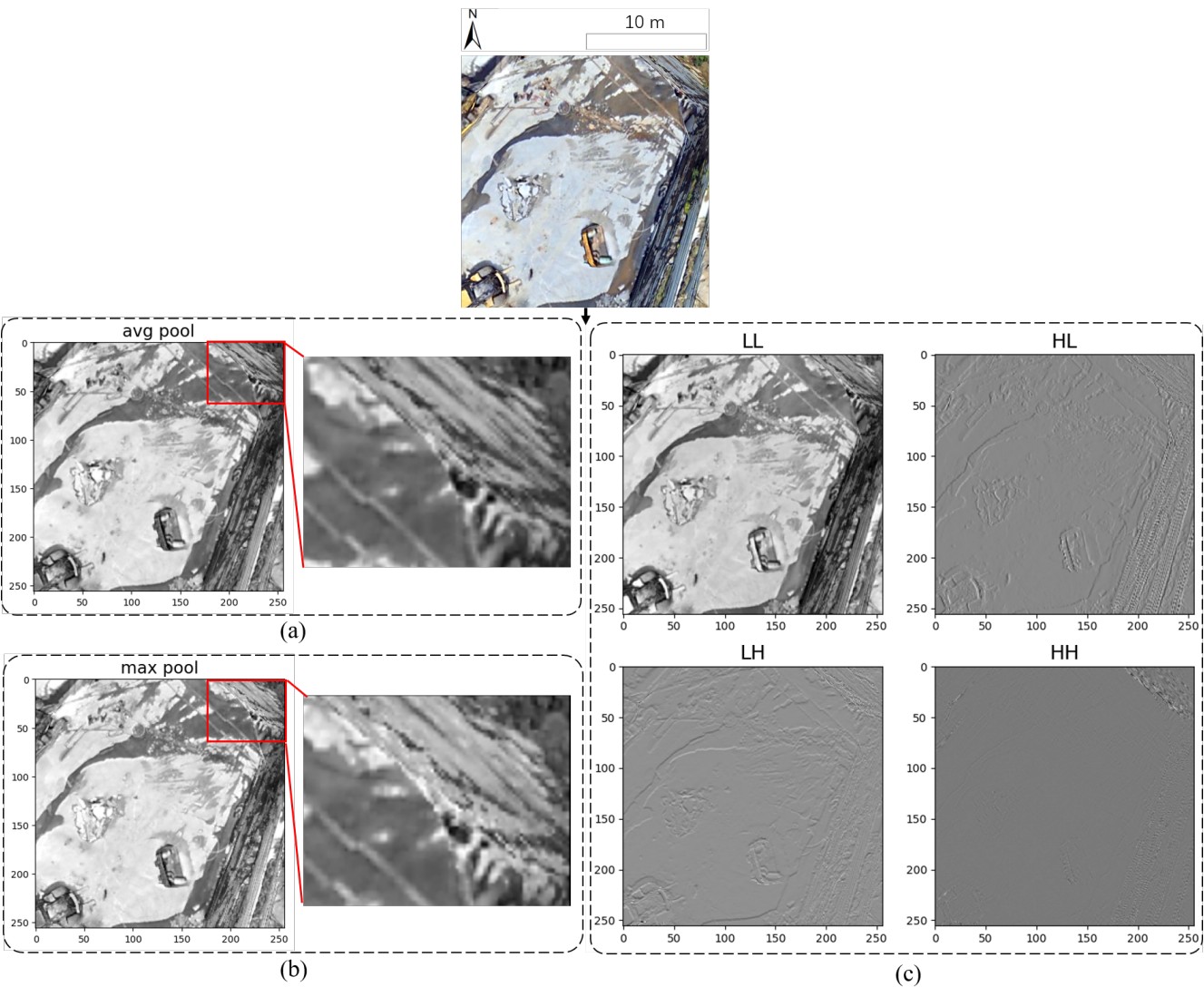

**Figure 12.** Visualization results of average pooling, max pooling and discrete wavelet transform on input image. (**a**) Average pooling; (**b**) max pooling; (**c**) discrete wavelet transform.

Furthermore, consideration is given to feature maps created by concatenating in the channel dimension and modifying the number of channels through convolution, which results in the creation of new feature information. It is crucial to not overlook this information, emerging from the fusion process. Such information should be seamlessly incorporated into feature maps at multiple levels, especially in the Perception process of the GIP module. Through this series of operations, the model's ability to discern feature information in input images is evaluated, utilizing Class Activation Mapping (CAM) [60]. The red areas'

presence and intensity in the CAM highlight the model's enhanced detection ability in those zones, as illustrated in Figure 13.

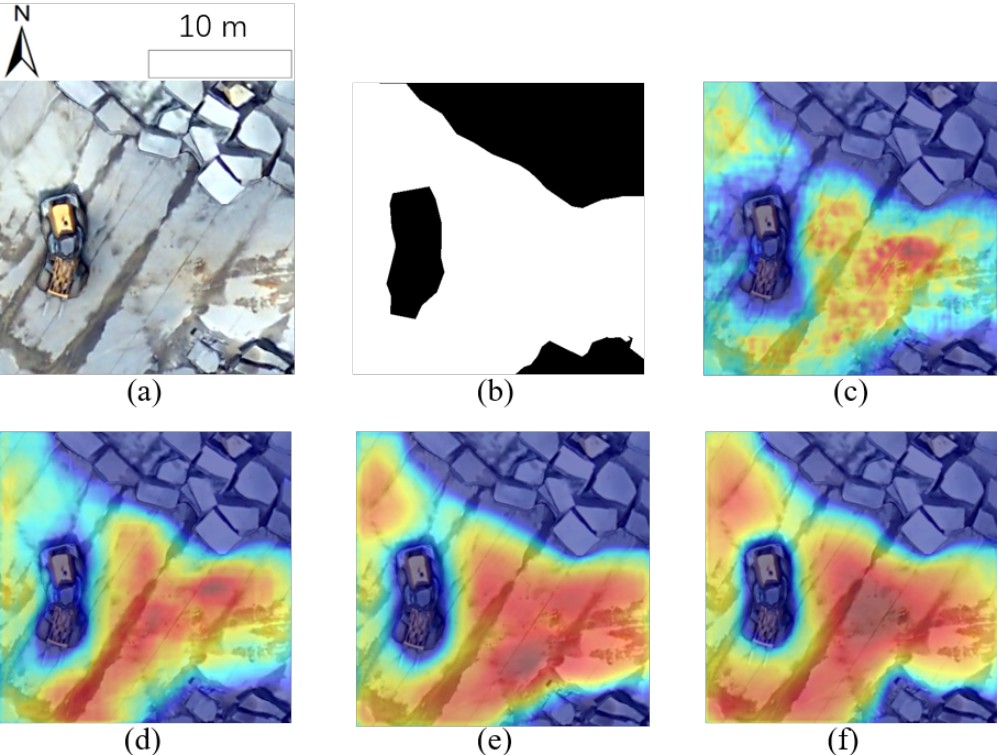

**Figure 13.** Visualization results of CAM on input image. (**a**) Image; (**b**) Ground Truth; (**c**) UNet; (**d**) UNet + FPN; (**e**) UNet + GD; (**f**) GIPNet.

*5.2. Limitation and Potential Improvements*

The utilization of drones in mining has significantly advanced data acquisition for open-pit mining area recognition, creating a robust dataset foundational for this research [19,61,62]. This study emphasizes the importance of identifying and outlining mining areas to formulate effective mining strategies and detect illegal mining activities. Techniques like oblique photogrammetry and 3D modeling are integrated to build a specialized dataset, focusing primarily on high-resolution drone imagery for recognizing open-pit granite mining areas. Despite the promising outcomes and the establishment of a novel research framework, there remains scope for ongoing refinement and exploration within this domain.

Challenges and future directions in this research include the time-intensive process of manual dataset annotation, essential for verifying the proposed technical methods. Given the surplus of images from drones beyond those used in this study, exploring weakly supervised and unsupervised deep learning for open-pit mining recognition presents a promising future trend. Furthermore, the limited size and scope of the dataset, confined to a single research area and specific data collection tools, indicate a need for diversifying data sources to enhance model scalability. Finally, leveraging the image dataset from the 3D modeling of mining sites, combined with geographical digital data, facilitates the production of two-dimensional segmentation maps. Integrating these with three-dimensional site representations could enable advanced analyses for volume calculation and excessive excavation detection in mining areas.

## 6. Conclusions

To obtain high-precision and timely information on open-pit mining areas is of great significance for the mining industry in carrying out production plans, preventing illegal

mining activities, and protecting the ecological environment. To achieve such results, high-quality image data are indispensable. Drones, as a low-cost, low-risk, and high-precision remote sensing technology, combined with rapidly developing deep learning methods, can fully leverage each other's advantages.

This study took the No. 2 granite mine in Xiling, Huashan Township, Zhongshan County, Hezhou City, Guangxi Province, as the study area, selecting 22 mining areas supported by drone orthophoto images as experimental data. Furthermore, a new GIPNet is designed to propose improvements from two perspectives: reducing information loss in multi-scale feature fusion and enhancing the boundary recognition ability of open-pit mining areas. The Gather–Injection–Perception (GIP) module divides multi-scale feature fusion into low-level and high-level fusion stages. By gathering feature maps of different scales to form global information, it injects them into feature maps of each scale. In this process, upsampling and downsampling are required for scale unification. In the low-level stage, discrete wavelet transform is used instead of ordinary downsampling to preserve more feature information. In the high one, a dual-branch attention mechanism is designed to distinguish high-frequency features from low-frequency features. Additionally, new perceptual pathways are proposed to further integrate multi-scale information. The Boundary Perception (BP) module, through the design of boundary aggregation and upsampling attention modules, better utilizes the high-dimensional semantic information and low-dimensional detail information output by the GIP module, improving the model's ability to recognize the boundaries of open-pit mining areas.

The proposed GIPNet demonstrates significant effectiveness, achieving 90.67% Precision, 92.00% Recall, 91.33% F1-score, and 84.04% IoU. These experimental results demonstrate competitiveness when compared with results from other classic and advanced network models. Ablation analysis proved the effectiveness of the GIP module and BP module. Moreover, the proposed framework is applicable to different backbones.

Future research will focus on the integration and application of multi-source image data, weakly supervised, and unsupervised learning to enhance the model's generalizability. Additionally, the integration of 2D segmentation and 3D volume calculation for mining areas is planned to be carried out, broadening the applications of drones and deep learning in the mining industry.

**Author Contributions:** Conceptualization, D.Z. and X.M.; methodology, D.Z. and S.D.; validation, D.Z.; formal analysis, D.Z., X.M. and S.D.; resources, S.D., X.M. and C.Y.; data curation, D.Z. and X.M.; writing—original draft preparation, D.Z.; writing—review and editing, S.D., X.M. and C.Y.; visualization, D.Z. and X.M.; supervision, X.M. and C.Y.; project administration, D.Z., X.M. and S.D.; funding acquisition, X.M. and C.Y. All authors have read and agreed to the published version of the manuscript.

**Funding:** This research was funded by the National Natural Science Foundation of China (NSFC) under Grant No. 41101417 and No. 41971352.

**Data Availability Statement:** Data is contained within the article.

**Acknowledgments:** The authors are very grateful to the many people who helped to comment on the article. Special thanks to the editors and reviewers for providing valuable insight into this article.

**Conflicts of Interest:** The authors declare no conflicts of interest.

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
