# Peer review of "Open-Pit Granite Mining Area Extraction Using UAV Aerial Images and the Novel GIPNet"

_remotesensing, doi:10.3390/rs16050789_

Round 1
Reviewer 1 Report
Comments and Suggestions for Authors
Hello to dear authors
Many thanks for your scientific efforts and development of new mining techniques. Your manuscript is a good technical research on remote sensing and its application in rock mining. There are some comments from me to improve the manuscript, please read them carefully and revise the corrections:
(1) Lines 4 & 5: Please change this sentence with another interpretation or remove it: We introduce an innovative network that significantly enhances the accuracy of mining area segmentation.
(2) Line 2: Your research is a new and creative method, but at the beginning of the abstract, you should talk about the general history in 2 sentences (2 lines).
(3) Line 13: This writing structure needs to be changed: Our method demonstrates ..., Please check and change throughout the manuscript. For example: our model -> suggested model.
(4) Line 19: Keywords: mining area extraction is not correct, please change it to -> Mining or Rock extraction or Quarry.
(5) Line 30 & 31: Unnecessary sentence, please remove it: Open-pit mining, a form of surface mining, involves the extraction of minerals after 30 removing and discarding surrounding rock. Please read this article https://www.mdpi.com/2075-163X/13/9/1160 to understand the concept of separation of rock boundaries and use it as a reference.
(6) Figure 1: Please remove this figure from the introduction, create a section entitled of Raw Data, and in addition to more explanations about the studied mine (name of the mine, exact type of rock, geographical location of the mine (it is better to mark it on the map)), and put figure 1 in this section.
(7) Lines 120 to 124: You need to clearly state the objectives of the research at the end of the introduction.
(8) Figure 1: All images in this figure must be to scale.
(9) Lines 128 to 132: Unnecessary sentences, please remove all of them.
(10) Line 133: Methodology: Please explain your technical follow of the research through a comprehensive follow chart (After a brief explanation about the methodology, this image will be presented.), Now the reader gets directly into the details of the method and confused.
(11) Figure 2: Line 133: You should present Figure 2 after you mention it in the text, not before mentioning.
(12) Figures of 9 and 10: You stated earlier that the background (where there is no mine boundary and is covered with soil or vegetation) is identified by the number 0 (black color in these images), the analyzes presented in these images in order to identify the boundaries indicate that In some places, machines or stones extracted on the floor of the mine are considered as background! What is your justification for this topic? And what answer does your method have for this problem? Has the result improved? What is your definition of the boundary in an open pit mine?
Best regards
Comments on the Quality of English LanguageAvoid the term "our method" or "our module". You can use the suggested method. Please check throughout the text.
Author Response
Thank you for your comments, we have uploaded the point-to-point responses, please check the attachment.

Reviewer 2 Report
Comments and Suggestions for Authors
The manuscript gives a method based on the BIPNet network for granite recognition in mining areas, but the method is not innovative enough, the improvement of the recognition using this method compared to other methods is not outstanding, and the example applied is very single. It is suggested to add more application examples.
Author Response

(The authors gave the same response as above.)

Reviewer 3 Report
Comments and Suggestions for Authors
This study utilizes unmanned aerial vehicle's (UAVs) derived orthophoto imaging for a local mining area in Zhongshan County, Hezhou City, Guangxi, China. The authors have developed the Gather Injection-Perception (GIP) workflow to preserve multi-scale feature information along with Boundary Perception (BP) module to improve the accuracy of blurred boundaries and geometric shapes of the open pit area areas. Based on the GIPNet model, they achieved 90.67% precision, 92.00% recall, 91.33% F1-score, and 84.04% IoU results. Overall, the manuscript is very well written, and the objectives are clearly set for the readers. I recommend this manuscript for publication, but please consider following points below:
Scales are missing in figure 1. Please insert the scale along with North direction. Also, insert scale and north direction in figures 6b, 8a, 9a, 10a and 13a.
Add image side and overlap ratios in subsection 3.1.
It would be good to divide section 1 (introduction) into two subsections, i.e., 1.1) background or previous work related to this study, and 1.2) Introduction to the GIPNet developed in this study.
76: References needed at the end of this sentence.
Figure 2 should come after the last para of subsection 2.1. Similarly, Figure 3 should come after line 182. Please check other figures with respect to the running text.
380: “This research investigates an open-pit granite mine located in Zhongshan County, Hezhou City, Guangxi, China”, This should already come in the introductory section as a case study.
Author Response

(The authors gave the same response as above.)

Reviewer 4 Report
Comments and Suggestions for Authors
It is a comprehensive study. However, besides the UAV Aerial Images and the Novel GIPNet, other UAV methods are applicable. I propose that authors must include on paragraph in Introduction describing other methods: for instance, Eppelbaum and Mishne (2011), Antoine et al. (2020).
Eppelbaum, L.V. and Mishne, A.R., 2011. Unmanned Airborne Magnetic and VLF investigations: Effective Geophysical Methodology of the Near Future. Positioning, 2, No. 3, 112-133.
Antoine, R. et al., 2020. Permeability and voids influence on the thermal signal, as inferred by multitemporal UAV-based infrared and visible images. Jour. of Hydrology, 587, 124907, 1-17.
Moderate editing of English language required. Please see the revised Conclusions below.
In this study, we explored using uncrewed aerial vehicles (UAVs) in the mining sector, focusing on creating a drone-captured image dataset for identifying open-pit granite mines. Using this dataset, our investigation evaluated the performance of various deep learning models in mining area segmentation and extraction. We identified limitations in traditional multi-scale feature fusion methods and introduced the Gather-Injection- Perception (GIP) module to preserve multi-scale information effectively. Additionally, the Version December 18, 2023, submitted to Journal Not Specified 23 of 25 Boundary Perception(BP) module, was developed to enhance the accuracy of geometric shape recognition in mining areas. Our proposed GIPNet model demonstrated significant effectiveness, achieving 90.67% Precision, 92.00% Recall, 91.33% F1-score, and 84.04% IoU. These results validate GIPNet’s competitive performance. Future research will focus on diversifying the angles and sources of image data for open-pit mining area recognition, employing weakly supervised and unsupervised learning to improve the model’s general-is ability. We also plan to integrate 2D segmentation and 3D volume calculation for mining areas, broadening the applications of drones and deep learning in the mining industry.
Comments on the Quality of English Language-
Author Response

(The authors gave the same response as above.)

Reviewer 5 Report
Comments and Suggestions for Authors
The article is devoted to Open-Pit Granite Mining Area Extraction Using UAV Aerial Images and the Novel GIPNet. The novel Gather-Injection-Perception (GIP) module is central to the presented approach. It is designed to overcome the information loss typically associated with conventional feature pyramid modules during feature fusion.
The theoretical background of this research is sufficient. The paper is presented clearly. However, there are some minor remarks to be mentioned.
(1) Some additional index terms must extend a list of Keywords.
(2) The Introduction should shed more light on the nuances of GIPNet (line 108) since you mention it.
(3) Fig.2: the system's description needs to be added in the following paragraphs. Besides, there is no explanation for the presented blocks like F, P, N, and LC.
(4) Lines 450-452: more details must be disseminated concerning the systems used for comparison.
(5) Some references ([5, 15, 18, 51]) lack essential information.
(6) Is sharing a public link to the source code possible?
(7) How many tunable parameters do you have in the GIPNet? It looks interesting to add another Table with different parameters. How long does it take to train the network?
Comments on the Quality of English LanguageThere are some minor issues, but nothing critical. The manuscript must be checked once again and polished.
Author Response

(The authors gave the same response as above.)

Round 2
Reviewer 1 Report
Comments and Suggestions for Authors
Hello to the dear authors
Based on the number of comments in the previous review report :
About Comment #(3):
In the response's report, it is said that similar structures have been checked and corrected throughout the text of the manuscript; But this is not so. For Example : Line 389 (Last version of the manuscript) : "we opted for a weighted binary cross-entropy...", Line 394: "The loss function we have developed...", Line 463: "we employed widely used evaluation metrics...". The subject is related to the use of the passive term of verbs and sentences in a scientific writing.
About Comment #(4):
I have mentioned "Mining" or "Rock (Stone) Extraction" but you wrote Mining Extraction ! It has no meaning in mining science. Please add Quarry to the keywords.
About Comment #(10):
I asked you to design a flow chart, but your answer was as follows: "However, we do not think it is necessary to add a new flow chart." I must say that this flowchart helps to read the article fluently. The current flowchart does not explain the methodological process, and I am presenting this as a reviewer of Remote Sensing Journal. Please read some of the published articles. The reader gets confused when faced with the current flowchart. It should be simple and clear. I am waiting for the comprehensive flowchart in the next responses.
About Comment #(12):
Thanks, the answer and the changes were justified.
New Comments:
(13) Introduction:
Dear authors, please take the comments of the reviewers seriously. The introduction is a solid text that consists of 3 or 4 paragraphs and has no subsections! Please read some related articles in Remote Sensing Journal to see what I mean. The introduction is currently unacceptable.
(14) Lines 490 to 532 are about methodology. You should not include them in the results section. As it is clear from the name of the section, you should only bring the results. Information about methods and methodology should all be provided in the methodology section. Also, writing in this way is quite confusing for the reader, remember that this manuscript will be read by everyone as an article. Please provide the information in the form of a summarized table in the methodology section.
Best wishes
Comments on the Quality of English LanguageBased on the number of comments in the previous review report :
About Comment #(3):
In the response's report, it is said that similar structures have been checked and corrected throughout the text of the manuscript; But this is not so. For Example : Line 389 (Last version of the manuscript) : "we opted for a weighted binary cross-entropy...", Line 394: "The loss function we have developed...", Line 463: "we employed widely used evaluation metrics...". The subject is related to the use of the passive term of verbs and sentences in a scientific writing.
Author Response
Thank you for your comments. We have revised the manuscript according to your comments and made a point-to-point response in the attachment. Please check the attachment.

Reviewer 2 Report
Comments and Suggestions for Authors
I recommend it is accepted.
Comments on the Quality of English LanguageAs it is.